# ACHIEVING MARGIN MAXIMIZATION EXPONENTIALLY FAST VIA PROGRESSIVE NORM RESCALING

## ABSTRACT

In this work, we investigate the margin-maximization bias exhibited by gradient-based algorithms in classifying linearly separable data. We present an in-depth analysis of the specific properties of the velocity field associated with (normalized) gradients, focusing on their role in margin maximization. Inspired by this analysis, we propose a novel algorithm called Progressive Rescaling Gradient Descent (PRGD) and show that PRGD can maximize the margin at an exponential rate. This stands in stark contrast to all existing algorithms, which maximize the margin at a slow polynomial rate. Notably, we identify mild conditions under which we show that existing algorithms such as gradient descent (GD) and normalized gradient descent (NGD) provably fail in maximizing the margin efficiently. To validate our theoretical findings, we present both synthetic and real-world experiments. Notably, PRGD also shows promise in enhancing the generalization performance when applied to linearly non-separable datasets and deep neural networks.

## 1 INTRODUCTION

In modern machine learning, models are often over-parameterized in the sense that they can easily interpolate training data, giving rise to a loss landscape with many global minima. Although all these minima yield zero training loss, their generalization ability can vary significantly. Intriguingly, it is often observed that Stochastic Gradient Descent (SGD) and its variants consistently converge to solutions with favorable generalization properties even without needing any explicit regularization (Neyshabur et al., 2014; Zhang et al., 2021). This phenomenon implies that the "implicit bias" inherent in SGD plays a crucial role in ensuring the efficacy of deep learning; therefore, revealing the underlying mechanism is of paramount importance.

Soudry et al. (2018) investigated this issue in the context of classifying linearly separable data with linear models. The study showed that gradient descent (GD) trained with exponentially-tailed loss functions can implicitly maximize the $\ell_2$-margin during its convergence process, ultimately locating a maximum-margin solution. This discovery offers valuable insights into the superior generalization performance often observed with GD, as larger margins are generally associated with improved generalization (Boser et al., 1992; Bartlett et al., 2017). However, the margin maximization rate of GD has been proven to be extremely slow, at a rate of $\mathcal{O}(1/\log t)$.

Since then, many researchers have dedicated themselves to designing algorithms aimed at accelerating the margin maximization in this problem. Notably, Nacson et al. (2019b); Ji & Telgarsky (2021) employ GD with an aggressive step size to improve this margin maximization rate, and Ji & Telgarsky (2021) demonstrated that GD with an aggressive step size can achieve polynomially fast margin maximization at a $\mathcal{O}(1/t)$ rate. More recently, Ji et al. (2021) introduced a momentum-based gradient method by applying Nesterov acceleration to the dual formulation of this problem. Their approach attains a remarkable margin maximization rate of $\tilde{\mathcal{O}}(1/t^2)$, currently standing as the state-of-the-art algorithm for this problem.

In this work, we present a systematic analysis of the unique properties of the velocity field related to (normalized) gradients, highlighting that the centripetal velocity is a key factor in determining the rate of margin maximization. Notably, we identify mild conditions, under which the above margin-maximization rates: $\mathcal{O}(1/t)$ for NGD and $\mathcal{O}(1/\log t)$ for GD are tight, explaining why GD and NGD are inefficient in maximizing the margin. This is due to the fact that the gradients tend to align closely with the direction of the regularization path, causing the centripetal velocity to diminish

during convergence. These insights inform a strategy to speed up the margin maximization via maintaining a non-degenerate centripetal velocity:

- We first show that there exists a favorable semi-cylindrical surface that is away from the regularization path and as such, the centripetal velocity is uniformly lower-bounded there. Leveraging this property, we introduce a novel algorithm called PRGD, which cyclically rescales parameters to semi-cylindrical surfaces with progressive radius. In order to keep the iterations on the semi-cylindrical surfaces, we perform projection in each step. Notably, we prove that PRGD can maximize the margin at an exponential rate $\mathcal{O}(e^{-\Omega(t)})$.

- We then validate our theoretical findings through both synthetic and real-world experiments. In particular, when applying PRGD to non-separable datasets and homogenized deep neural networks—beyond the scope of our theory—we still observe consistent test performance improvements. This suggests that our theory can be potentially extended to nonlinear homogenized networks.

We summarize our theoretical results and the comparison with existing ones in Table 1.

Table 1: Comparison of the directional convergence rates of different algorithms under Assumption 3.1, 5.4, and $\boldsymbol{w}^\star \neq \frac{1}{|\mathcal{I}|} \sum_{i \in \mathcal{I}} \boldsymbol{x}_i y_i$.

| Algorithm | Directional Convergence Rate $\mathrm{e}(t) = \left\| \frac{\boldsymbol{w}(t)}{\|\boldsymbol{w}(t)\|} - \boldsymbol{w}^\star \right\|$ |
|---|---|
| GD | $\mathrm{e}(t) = \mathcal{O}(1/\log t)$ (Soudry et al., 2018), $\mathrm{e}(t_k) = \Theta(1/\log t_k)$ (Thm 6.4) |
| NGD | $\mathrm{e}(t) = \mathcal{O}(1/t)$ (Ji & Telgarsky, 2021), $\mathrm{e}(t_k) = \Theta(1/t_k)$ (Thm 6.4) |
| Dual Acceleration | $\mathrm{e}(t) = \mathcal{O}(1/t^2)$ (Ji et al., 2021) |
| PRGD | $\mathrm{e}(t) = e^{-\Omega(t)}$ (Thm 6.2) |

## 2 RELATED WORK

Understanding the implicit bias of optimization algorithms is one of the most important problems in deep learning theory. This topic has been extensively studied recently, and in this section, we only review those that are closely related to the current work.

**Margin maximization of GD.** The margin-maximization bias of GD trained with exponentially-tailed loss functions was originally studied in Soudry et al. (2018). Except for works mentioned above, Ji & Telgarsky (2018b) investigated the margin-maximization bias of GD for classifying datasets that are not linearly separable. Nacson et al. (2019c) proved margin maximization for SGD. Furthermore, Gunasekar et al. (2018a); Wang et al. (2022); Sun et al. (2022) characterized the implicit bias of many other optimization algorithms. Recently, Wu et al. (2023) analyzed the impact of edge of stability (Cohen et al., 2021; Wu et al., 2018) for achieving margin maximization. In a related study, Ji et al. (2020) examined other types of loss functions and regularization path.

In a similar setup, researchers also explored the implicit bias of GD on nonlinear models, such as deep neural networks (DNNs). Specifically, Ji & Telgarsky (2018a); Gunasekar et al. (2018b) investigated the implicit bias on deep linear fully-connected and convolutional networks. Nacson et al. (2019a); Lyu & Li (2019); Ji & Telgarsky (2020) proved that GD on homogeneous DNNs converges to the KKT direction of an $\ell_2$ max-margin problem. Recently, Kunin et al. (2023) extended this result to quasi-homogeneous networks.

**Other Implicit biases.** It is widely believed that flatter minima lead to better generalization (Hochreiter & Schmidhuber, 1997; Keskar et al., 2016). Recent studies (Wu et al., 2018; Ma & Ying, 2021; Wu et al., 2022) provided explanations for why SGD tends to select flat minima on DNNs, using dynamical stability analysis. Additionally, Woodworth et al. (2020); Nacson et al. (2022) investigated how initialization scale and step size affect the selection bias of GD between the "kernel" and "rich" regime on linear diagonal neural networks.

## 3 PRELIMINARIES

**Notation.** We use bold letters for vectors and lowercase letters for scalars, e.g. $\boldsymbol{x} = (x_1, \cdots, x_d)^\top \in \mathbb{R}^d$. For any vector $\boldsymbol{v}$, we use $\hat{\boldsymbol{v}} = \boldsymbol{v}/\|\boldsymbol{v}\|$ the normalized vector. We use $\langle \cdot, \cdot \rangle$ for the standard

Euclidean inner product between two vectors, and $\|\cdot\|$ for the $\ell_2$ norm of a vector or the spectral norm of a matrix. We use standard big-O notations $\mathcal{O}, \Omega, \Theta$ to hide absolute positive constants, and use $\tilde{\mathcal{O}}, \tilde{\Omega}, \tilde{\Theta}$ to further hide logarithmic constants. For any positive integer $n$, let $[n] = \{1, \cdots, n\}$.

**Classification Problem.** In this paper, we consider the binary classification problem. Given a dataset $\mathcal{S} = \{(\boldsymbol{x}_1, y_1), \cdots, (\boldsymbol{x}_n, y_n)\}_{i=1}^n \subset \mathbb{R}^d \times \{\pm 1\}$, we need to find some model $f(\cdot; \boldsymbol{\theta}) : \mathbb{R}^d \to \mathbb{R}$ to classify all data correctly, .i.e., $y_i f(\boldsymbol{x}_i; \boldsymbol{\theta}) > 0$. Without loss of generality, we assume $\|\boldsymbol{x}_i\| \leq 1$ for all $i \in [n]$. Moreover, we consider the following linearly separable data, which is a standard assumption in analyzing the implicit bias of GD (Soudry et al., 2018; Nacson et al., 2019b; Ji & Telgarsky, 2021)

**Assumption 3.1** (linear separability). *There exists $\boldsymbol{w} \in \mathbb{S}^{d-1}$ such that $\min_{i \in [n]} y_i \langle \boldsymbol{w}, \boldsymbol{x}_i \rangle > 0$.*

A classical method to solve the binary classification problem is the $\ell_2$ Support Vector Machine ($\ell_2$-SVM), which need to solve the optimization problem: $\max_{\|\boldsymbol{w}\| \leq 1} \min_{i \in [n]} y_i \langle \boldsymbol{w}, \boldsymbol{x}_i \rangle$. Since $\ell_2$-SVM is equivalent to a strongly convex quadratic programming problem with linear constraints, we have the following classical result: under Assumption 3.1, the $\ell_2$ SVM problem has a *unique* optimal solution $\boldsymbol{w}^\star \in \mathbb{S}^{d-1}$.

**Margin and Max-margin.** Consequently, under Assumption 3.1, we can define the margin of $\boldsymbol{w} \in \mathbb{R}^d$ as $\gamma(\boldsymbol{w}) := \min_{i \in [n]} y_i \left\langle \frac{\boldsymbol{w}}{\|\boldsymbol{w}\|}, \boldsymbol{x}_i \right\rangle$. Moreover, we denote the max-margin and max-margin direction as $\gamma^\star := \max_{\|\boldsymbol{w}\| \leq 1} \min_{i \in [n]} y_i \langle \boldsymbol{w}, \boldsymbol{x}_i \rangle$ and $\boldsymbol{w}^\star := \arg\max_{\|\boldsymbol{w}\| \leq 1} \min_{i \in [n]} y_i \langle \boldsymbol{w}, \boldsymbol{x}_i \rangle$, respectively.

**Logistic Regression.** Another classical machine learning algorithm to solve the binary classification problem is the following logistic regression:

$$\min_{\boldsymbol{w} \in \mathbb{R}^d} \mathcal{L}(\boldsymbol{w}) = \frac{1}{n} \sum_{i=1}^n \ell\left(-y_i \langle \boldsymbol{w}, \boldsymbol{x}_i \rangle\right). \tag{1}$$

where $\ell(\cdot) : \mathbb{R} \to \mathbb{R}$ is an exponential-type loss function (Soudry et al., 2018; Nacson et al., 2019b). This includes widely-used classification loss functions such as the exponential loss and logistic loss. For the sake of simplicity, our analysis will focus on the exponential loss $\ell(z) = e^{-z}$, although it can be easily extended to the logistic loss $\ell(z) = \log(1 + e^{-z})$.

As a baseline algorithm, GD can be implied to solve Problem (1).

$$\textbf{GD:} \qquad \boldsymbol{w}(t+1) = \boldsymbol{w}(t) - \eta \nabla \mathcal{L}(\boldsymbol{w}(t)), \tag{2}$$

Soudry et al. (2018) showed under Assumption 3.1, GD (2) with $\eta \leq 1$ converges to the $\ell_2$ max-margin solution while minimizing the loss. However, this occurs at a slow rate $\gamma^\star - \gamma(\boldsymbol{w}(t)) = \mathcal{O}(1/\log t)$. To enhance this implicit bias, one can adopt the following Normalized Gradient Descent (NGD) with $\eta \leq 1$ (GD with an aggressive step size) to achieve margin maximization at a polynomial rate $\gamma^\star - \gamma(\boldsymbol{w}(t)) = \mathcal{O}(1/t)$ (Ji & Telgarsky, 2021). The update rule of NGD is:

$$\textbf{NGD:} \qquad \boldsymbol{w}(t+1) = \boldsymbol{w}(t) - \eta \frac{\nabla \mathcal{L}(\boldsymbol{w}(t))}{\mathcal{L}(\boldsymbol{w}(t))}. \tag{3}$$

**Regularization Path.** Our subsequent analysis will leverage the properties of regularization path (Hastie et al., 2004; Ji et al., 2020). Consider the regularized solution defined by $\boldsymbol{w}^\star_{\text{reg}}(B) := \arg\min_{\|\boldsymbol{w}\|_2 \leq B} \mathcal{L}(\boldsymbol{w})$. Then, the regularization path refers to the curve traced by $\boldsymbol{w}^\star_{\text{reg}}(\cdot)$ as $B$ varies, formally given by $\{\boldsymbol{w}^\star_{\text{reg}}(B)\}_{B > 0}$.

## 4 MOTIVATIONS AND THE ALGORITHM

In this section, we introduce our proposed algorithm and explain the motivation behind through two toy examples. First, we state some of our **key observations** of the structure of for the problem (1):

- **Homogeneity.** For the linear model $f(\boldsymbol{x}; \boldsymbol{w}) = \langle \boldsymbol{w}, \boldsymbol{x} \rangle$, **rescaling** the parameter $\boldsymbol{w}$ does not change the margin, i.e., $\gamma(\boldsymbol{w}) = \gamma(c\boldsymbol{w})$ for any $c > 0$ and $\boldsymbol{w} \in \mathbb{R}^d$.

- **Directional Convergence.** Under Assumption 3.1, it holds that $\gamma^\star - \gamma(\boldsymbol{w}) \leq \|\hat{\boldsymbol{w}} - \boldsymbol{w}^\star\|$ (Lemma A.4). This implies that the margin maximization rate can be controlled by the rate of directional convergence.

- **Convexity.** The Hessian is $\nabla^2 \mathcal{L}(\boldsymbol{w}) = \frac{1}{n} \sum_{i=1}^n e^{-\langle \boldsymbol{w}, \boldsymbol{x}_i y_i \rangle} \boldsymbol{x}_i \boldsymbol{x}_i^\top$. If all data has been classified correctly, i.e., $\min_{i \in [n]} \langle \boldsymbol{w}, \boldsymbol{x}_i y_i \rangle > 0$, then the convexity of the loss landscape is stronger in regions with smaller norm.

- **Centripetal Velocity.** Intuitively, if the descent direction $-\nabla \mathcal{L}(\boldsymbol{w})/\mathcal{L}(\boldsymbol{w})$ at some $\boldsymbol{w} \in \mathbb{R}^d$ has larger "centripetal" component (orthogonal to $\boldsymbol{w}^\star$), it will make more effective progress on the directional convergence towards $\boldsymbol{w}^\star$. Furthermore, notice that for lots of datasets, the centripetal velocity is greater at the points farther from the regularized path, which we will explain in detail below using Dataset 4.1 as an example.

Following the above observations, we can conclude

- On the one hand, in order to obtain **greater centripetal velocity** for faster directional convergence, we should rescale the parameter $\boldsymbol{w} \to c\boldsymbol{w}$ ($c > 1$) sufficiently away from the regularized path. In Algorithm 1, this point corresponds to the progressive scaling steps.

- On the other hand, the landscape convexity in small-norm region is stronger than that in large-norm region. Therefore, one should accelerate the local optimization by taking as many steps as possible by using small $\boldsymbol{w}$. In Algorithm 1, this point corresponds to the projected GD steps.

By combining the above two intuitions, we propose the Progressive Projected Gradient Descent (PPGD) in Algorithm 1.

---

**Algorithm 1:** Progressive Rescaling Gradient Descent (PRGD)

---

**Input:** Dataset $\mathcal{S}$; Initialization $\boldsymbol{w}(0)$; Progressive Time $\{T_k\}_{k=0}^K$; Progressive Radius $\{R_k\}_{k=0}^K$;

**for** $k = 0, 1, 2, \cdots, K$ **do**

    $\boldsymbol{w}(T_k + 1) = R_k \frac{\boldsymbol{w}(T_k)}{\|\boldsymbol{w}(T_k)\|}$;

    **for** $T_k + 1 \leq t \leq T_{k+1} - 1$ **do**

        $\boldsymbol{v}(t+1) = \boldsymbol{w}(t) - \eta \frac{\nabla \mathcal{L}(\boldsymbol{w}(t))}{\mathcal{L}(\boldsymbol{w}(t))}$;

        $\boldsymbol{w}(t+1) = \mathrm{Proj}_{\mathbb{B}(\boldsymbol{0}, R_k)}(\boldsymbol{v}(t+1))$;

**Output:** $\boldsymbol{w}(T_K + 1)$.

---

Next, we substantiate the above observations and explain the mechanisms by which PRGD works via the toy problem:

**Dataset 1.** $\mathcal{S} = \{(\boldsymbol{x}_1, y_1), (\boldsymbol{x}_2, y_2), (\boldsymbol{x}_3, y_3)\}$ where $\boldsymbol{x}_1 = (\sqrt{1 - \gamma^{\star 2}}, \gamma^\star)^\top$, $y_1 = 1$, $\boldsymbol{x}_2 = (-\sqrt{1 - \gamma^{\star 2}}, \gamma^\star)^\top$, $y_2 = 1$, $\boldsymbol{x}_3 = (\sqrt{1 - \gamma^{\star 2}}, -\gamma^\star)^\top$, $y_3 = -1$, and $\gamma^\star > 0$ is small enough.

For this dataset, we have the following (tight) margin maximization and directional convergence results for both NGD and PRGD.

**Proposition 4.1.** *Consider Dataset 1. Then NGD (3) can only maximize the margin polynomially fast, while PRGD (Alg 1) can maximize the margin exponentially fast. Specifically,*

*(I) Let $\boldsymbol{w}(t)$ be NGD (3) solution at time $t$ with $\eta = 1$ starting from $\boldsymbol{w}(0) = \boldsymbol{0}$. Then both the margin maximization and directional convergence are at (tight) **polynomial** rates:*

$$\left\| \frac{\boldsymbol{w}(t)}{\|\boldsymbol{w}(t)\|} - \boldsymbol{w}^\star \right\| = \Theta(1/t), \quad \gamma^\star - \gamma(\boldsymbol{w}(t)) = \Theta(1/t);$$

*(II) Let $\boldsymbol{w}(t)$ be the PRGD solution (Algorithm 1) with $\eta = 1$ starting from $\boldsymbol{w}(0) = \boldsymbol{0}$. If we choose $R_k = e^{\Theta(k)}$ and $T_k = \Theta(k)$, then both the margin maximization and directional convergence are at (tight) **exponential** rate:*

$$\left\| \frac{\boldsymbol{w}(t)}{\|\boldsymbol{w}(t)\|} - \boldsymbol{w}^\star \right\| = e^{-\Theta(t)}, \quad \gamma^\star - \gamma(\boldsymbol{w}(t)) = e^{-\Theta(t)}.$$

Proposition 4.1 provides a comparative analysis of the efficiency of PRGD and the challenges faced by NGD. Next, we provide an intuitive explanation and a brief outline of the proof and the complete proof is available in Appendix A.

For this dataset, the max-margin direction is $\boldsymbol{w}^\star = (0,1)^\top$ and the regularized path satisfies $\lim_{R\to\infty} w^\star_{\mathrm{reg},1}(R) = -\frac{\log 2}{2\sqrt{1-\gamma^{\star 2}}}$. And in Figure 1, we plot the asymptotic line of the regularization path (the green curve) and the max-margin direction $\boldsymbol{w}^*$ (the red curve) for this dataset. Notably, the two lines are parallel with each other.

**Centripetal Velocity.** In this example, the centripetal velocity (orthogonal to $\boldsymbol{w}^\star$) is $[-\nabla\mathcal{L}(\boldsymbol{w})/\mathcal{L}(\boldsymbol{w})]_1 \mathrm{sgn}(w_1)$. As shown in Figure 1, one can see that the centripetal velocity is significant for $\boldsymbol{w}$ far away from the regularized path (outside the green zone), while the centripetal velocity in the green zone is tiny.

**Inefficiency of NGD.** As shown in Figure 1, the trajectory of NGD (the orange line) always remains near the regularized path in which $-\nabla\mathcal{L}(\boldsymbol{w})/\mathcal{L}(\boldsymbol{w})$ is nearly parallel to $\boldsymbol{w}^\star$ and the centripetal component (along $\boldsymbol{e}_1$) there is very small. Actually, we can show that NGD always stays in the green zone $\mathbb{A} := \left\{\boldsymbol{w}: w_1 \in \left[-3\log 2/4\sqrt{1-\gamma^2}, -\log 2/4\sqrt{1-\gamma^2}\right]\right\}$. Since the norm grows at $\Theta(t)$ rate (Lemma C.3), NGD is cursed to have only $\Theta(1/t)$ directional convergence rate.

**Efficiency of PRGD.** We consider a simple hyperparameter selection situation, $T_{k+1} - T_k = 2$, that is, do one step of progressive scaling and one step of projected normalized gradient descent in each period. As Fig 1 shown, PRGD (the purple line) can just solve the hardness that NGD is trapped in the green zone $\mathbb{A}$ with small centripetal velocity. The stretching step can ensure that PRGD escapes from $\mathbb{A}$ and arrives in $w_1 = -1$, where the centripetal velocity is significant; then the projected gradient step can achieve use this significant centripetal velocity to make progress on the directional convergence. Moreover, the centripetal velocity in $\{\boldsymbol{w}: w_1 = -1\}$ has a uniformly positive lower bound, one can use this to prove that the directional convergence is exponentially fast via simple geometric calculation.

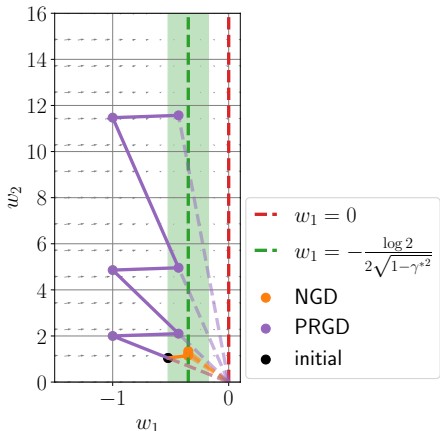

Figure 1: The vector field and the trajectories of NGD and PRGD for Dataset 1. The gray arrows plot the vector field $-\nabla\mathcal{L}(\cdot)/\mathcal{L}(\cdot)$. The red line corresponds to the max-margin direction, and the green area is around the regularized path. We run and visualize the trajectories of PPGD (purple) and NGD (orange) for 6 steps starting from the same initial point (black).

## 5 CENTRIPETAL VELOCITY ANALYSIS

Motivated by our analysis for Dataset 1, we formally study the angular velocity in this section. Moreover, inspired by the proof and visualization, we actually only need to focus on the angular velocity on an infinitely long semi-cylindrical surface in high-dimensional setting.

First, we give the following definition, which helps us decompose the parameters on $\mathbb{R}^d$ into essential directions.

**Definition 5.1** (Orthogonal Projection). we denote the projections of $\boldsymbol{w} \in \mathbb{R}^d$ along the direction $\boldsymbol{w}^\star$ and onto the orthogonal space of $\boldsymbol{w}^\star$ as $\mathcal{P}(\boldsymbol{w}) := \langle\boldsymbol{w}, \boldsymbol{w}^\star\rangle \boldsymbol{w}^\star$ and $\mathcal{P}_\perp(\boldsymbol{w}) := \boldsymbol{w} - \langle\boldsymbol{w}, \boldsymbol{w}^\star\rangle \boldsymbol{w}^\star$, respectively. It is worth noting that $\mathcal{P}(\boldsymbol{w}) + \mathcal{P}_\perp(\boldsymbol{w}) = \boldsymbol{w}$ holds for any $\boldsymbol{w} \in \mathbb{R}^d$.

Using this orthogonal projection, we can establish a formal definition for the "centripetal velocity".

**Definition 5.2** (Centripetal Velocity). The normalized gradient at $\boldsymbol{w} \in \mathbb{R}^d$ is $\nabla\mathcal{L}(\boldsymbol{w})/\mathcal{L}(\boldsymbol{w})$, and we define the centripetal velocity $\varphi(\boldsymbol{w})$ (towards $\boldsymbol{w}^\star$) at $\boldsymbol{w}$ by

$$\varphi(\boldsymbol{w}) := \left\langle -\frac{\nabla\mathcal{L}(\boldsymbol{w})}{\mathcal{L}(\boldsymbol{w})}, -\frac{\mathcal{P}_\perp(\boldsymbol{w})}{\|\mathcal{P}_\perp(\boldsymbol{w})\|}\right\rangle = \left\langle \frac{\nabla\mathcal{L}(\boldsymbol{w})}{\mathcal{L}(\boldsymbol{w})}, \frac{\mathcal{P}_\perp(\boldsymbol{w})}{\|\mathcal{P}_\perp(\boldsymbol{w})\|}\right\rangle.$$

Then we introduce the definition of infinitely long semi-cylindrical surface, which is the crucial geometry in our subsequent analysis.

**Definition 5.3** (Semi-cylindrical Surface). We use

$$\mathbb{C}(D; H) := \{\boldsymbol{w} \in \mathrm{span}\{\boldsymbol{x}_i : i \in [n]\} : \|\mathcal{P}_\perp(\boldsymbol{w})\| = D; \langle \boldsymbol{w}, \boldsymbol{w}^\star \rangle \geq H\}$$

to denote the infinitely long semi-cylindrical surface with the central direction $\boldsymbol{w}^\star$, the radius $D > 0$, and starting height $H > 0$.

Our subsequent analysis will concentrate on the semi-cylindrical surface as PRGD ensures the iterations will be confined in the surface. This surface is defined by its central direction, denoted by $\boldsymbol{w}^\star$, a radius $D > 0$, and extends infinitely in the direction of $\boldsymbol{w}^\star$ starting from a height $H$. Additionally, it is crucial to note that our attention is restricted to the smaller subspace $\mathrm{span}\boldsymbol{x}_i : i \in [n]$, rather than the entire space $\mathbb{R}^d$. This is justified by the observation that the trajectories of GD, NGD, and PRGD, when initialized from $\boldsymbol{0}$, will remain confined within this subspace indefinitely.

## 5.1 THEORETICAL ANALYSIS

In this subsection, we undertake a theoretical examination of the centripetal velocity, as defined in Definition 5.2, on the semi-cylindrical surface described in Definition 5.3. Our investigation aims to address the following query:

*Does a "favorable" semi-cylindrical surface exist where the centripetal velocity consistently maintains a positive lower bound?*

We demonstrate that such a favorable semi-cylindrical surface indeed exists, provided that the data are non-degenerate to a modest extent.

**Assumption 5.4** (Non-degenerate data (Soudry et al., 2018; Wu et al., 2023)). Let $\mathcal{I}$ be the index set of the support vectors, i.e., there exist $\alpha_i > 0$ ($i \in \mathcal{I}$) such that $\boldsymbol{w}^\star = \sum_{i \in \mathcal{I}} \alpha_i y_i \boldsymbol{x}_i$. We assume $\mathrm{span}\{\boldsymbol{x}_i : i \in \mathcal{I}\} = \mathrm{span}\{\boldsymbol{x}_i : i \in [n]\}$.

We remark Assumption 5.4 is widely used in prior implicit bias analysis, such as Theorem 4.4 in (Soudry et al., 2018) and (Wu et al., 2023).

Now we can state our main results about the centripetal velocity analysis.

**Theorem 5.5** (Centripetal Velocity Analysis, Main result). *Under Assumption 3.1 and 5.4, there exists a semi-cylindrical surface $\mathbb{C}(D; H)$ and a positive constant $\mu > 0$ such that*

$$\inf_{\boldsymbol{w} \in \mathbb{C}(D;H)} \varphi(\boldsymbol{w}) \geq \mu.$$

Theorem 5.5 establishes that for linearly separable and slightly non-degenerate dataset, there indeed exists a "good" semi-cylindrical surface in which the centripetal velocity has a uniformly positive lower bound. That is to say, on this semi-cylindrical surface, the negative normalized gradient has a *significance component* orthogonal to $\boldsymbol{w}^\star$ consistently. The proof is deferred to Appendix B.

# 6 MARGIN MAXIMIZATION AND DIRECTIONAL CONVERGENCE RATE

## 6.1 EXPONENTIAL FAST MARGIN MAXIMIZATION VIA PRGD

We have identified the condition ensuring the existence of the "good" semi-cylindrical surface, where the centripetal velocity is uniformly lower-bounded. For simplicity, in this section, we set this result as an assumption.

**Assumption 6.1.** There exists a semi-cylindrical surface $\mathbb{C}(D; H)$ and a positive constant $\mu > 0$ such that $\inf_{\boldsymbol{w} \in \mathbb{C}(D;H)} \varphi(\boldsymbol{w}) \geq \mu$.

The subsequent theorem shows that by leveraging our PRGD (Alg 1) and under the above assumption, the rate of directional convergence–and consequently, margin maximization–can be boosted to be exponential.

**Theorem 6.2** (PRGD, Main Result). *Under Assumption 3.1 and 6.1, let $\boldsymbol{w}(t)$ be solutions generated by the following **two-phase algorithms** starting from $\boldsymbol{w}(0) = \boldsymbol{0}$:*

- *Warm-up Phase: Run GD (2) with $\eta = 1/2$ for $T_w = \Theta(1)$ steps starting from $\boldsymbol{w}(0)$;*

- *Acceleration Phase: Run PRGD (Alg 1) with $\eta = \Theta(1)$, $R_k = e^{\Theta(k)}$ and $T_k = \Theta(k)$ starting from $\boldsymbol{w}(T_w)$.*

*Then, both directional convergence and margin maximization are achieved at **exponential** rate:*

$$\left\| \frac{\boldsymbol{w}(t)}{\|\boldsymbol{w}(t)\|} - \boldsymbol{w}^\star \right\| \le e^{-\Omega(t)}; \gamma^\star - \gamma(\boldsymbol{w}(t)) \le e^{-\Omega(t)}.$$

The complete proof is deferred to Appendix C and here, we provide a sketch of the proof to illustrate the intuition behind:

- In the warm-up phase, we employ GD to achieve a preliminary (slow) directional convergence, satisfying $\left\| \frac{\boldsymbol{w}(T_w)}{\|\boldsymbol{w}(T_w)\|} - \boldsymbol{w}^\star \right\| < \min\{D/2H, 1/2\}$, which is prepared to rescale $\boldsymbol{w}$ to the good semi-cylindrical surface $\mathbb{C}(D; H)$.
- After the warm up, we can rescale $\boldsymbol{w}(T_w)$ to the good semi-cylindrical surface $\mathbb{C}(D; H)$ by setting $R_1 = \frac{D}{\|\mathcal{P}_\perp(\boldsymbol{w}(T_w))\|}$. Then, applying projected gradient descent there can significantly speed up the directional convergence since the centripetal velocity on $\mathbb{C}(D; H)$ is well lower-bounded (Assumption 6.1). Then, by selecting suitable progressive scaling $R_k$, we can reposition the parameter back to $\mathbb{C}(D; H)$ again. Repeating this process, we will get effective directional convergence in each cycle. Finally, by simple geometric calculation, it can be proven that such directional convergence is exponentially fast.

Notice that in Proposition 4.1, we have provided a tightly exponentially fast rate on Dataset 1, which satisfies Assumption 3.1 and 6.1, hence, the tightness of Theorem 6.2 can be ensured.

**Corollary 6.3** (PRGD, non-degenerate dataset). *Under Assumption 3.1 and 5.4, let $\boldsymbol{w}(t)$ be solutions generated by the following **two-phase algorithms** starting from $\boldsymbol{w}(0) = \boldsymbol{0}$:*

- *Warm-up Phase: Run GD (2) or NGD (3) with $\eta \le 1$ for $T_w$ steps starting from $\boldsymbol{w}(0)$;;*

- *Acceleration Phase: Run PRGD (Alg 1) with $\eta = \Theta(1)$, $R_k = e^{\Theta(k)}$ and $T_k = \Theta(k)$ starting from $\boldsymbol{w}(T_w)$.*

*Then, both directional convergence and margin maximization are achieved at **exponential** rate:*

$$\left\| \frac{\boldsymbol{w}(t)}{\|\boldsymbol{w}(t)\|} - \boldsymbol{w}^\star \right\| \le e^{-\Omega(t)}; \quad \gamma^\star - \gamma(\boldsymbol{w}(t)) \le e^{-\Omega(t)}.$$

It is worth noting that Assumption 5.4 can imply Assumption 6.1. Therefore, Theorem 6.2 implies Theorem 6.3 with GD (Phase I) + PRGD (Phase II) directly. Additionally, a slight difference is that in Theorem 6.3, we can use NGD in Phase I (to obtain faster directional warm-up than GD), because Assumption 5.4 can further guarantee the directional convergence of NGD (Ji & Telgarsky, 2021).

## 6.2 INEFFICIENCY OF GD AND NGD

**Theorem 6.4** (GD and NGD, Main results). *Suppose Assumption 3.1 and 5.4 hold. Additionally, we assume $\gamma^\star \boldsymbol{w}^\star \ne \frac{1}{|\mathcal{I}|} \sum_{i \in \mathcal{I}} y_i \boldsymbol{x}_i$.*

- *For NGD (3) with $\eta \le \eta_0$ starting from $\boldsymbol{w}(0) = \boldsymbol{0}$ (where $\eta_0$ is a constant), we have there exists a subsequence $\boldsymbol{w}(t_k)$ ($t_k \to \infty$) such that $\left\| \frac{\boldsymbol{w}(t_k)}{\|\boldsymbol{w}(t_k)\|} - \boldsymbol{w}^\star \right\| = \Theta(1/t_k)$.*

- *For GD (2) with $\eta \le \eta_0$ starting from $\boldsymbol{w}(0) = \boldsymbol{0}$ (where $\eta_0$ is a constant), there exists a subsequence $\boldsymbol{w}(t_k)$ ($t_k \to \infty$) such that $\left\| \frac{\boldsymbol{w}(t_k)}{\|\boldsymbol{w}(t_k)\|} - \boldsymbol{w}^\star \right\| = \Theta(1/\log t_k)$.*

As presented in Table 1, under the same conditions–Assumption 3.1, 5.4, and $\gamma^\star \boldsymbol{w}^\star \ne \frac{1}{|\mathcal{I}|} \sum_{i \in \mathcal{I}} \boldsymbol{x}_i y_i$, PRGD can achieve directional convergence exponentially fast with the rate $\left\| \frac{\boldsymbol{w}(t)}{\|\boldsymbol{w}(t)\|} - \boldsymbol{w}^\star \right\| = e^{-\Omega(t)}$.

In contrast, Theorem 6.4 ensures that NGD maintains a tight bound of polynomial speed, and GD exhibits a tight bound with exponentially slow rate.

The detailed proof of Theorem 6.4 is provided in Appendix C. Although this proof is more complicate than Proposition 4.1 due to the more general dataset, their proof insights are highly similar. In this proof, we still focus on the dynamics of $\mathcal{P}_\perp(\boldsymbol{w}(t))$. Actually, we can prove that there exists a subsequence $\mathcal{P}_\perp(\boldsymbol{w}(t_k))$ $(t_k \to \infty)$, which convergences to some $\boldsymbol{v} \in \text{span}\{\mathcal{P}_\perp(\boldsymbol{x}_i) : i \in \mathcal{I}\}$. Moreover, our condition $\gamma^\star \boldsymbol{w}^\star \neq \frac{1}{|\mathcal{I}|}\sum_{i\in\mathcal{I}} \boldsymbol{x}_i y_i$ can ensure that $\boldsymbol{v}^\star \neq \boldsymbol{0}$. Therefore, $\|\mathcal{P}_\perp(\boldsymbol{w}(t_k))\| = \Theta(1)$. Since the norm grows at $\|\boldsymbol{w}(t_k)\| = \Theta(t_k)$ (Lemma C.3), NGD must have only $\Theta(1/t)$ directional convergence rate.

# 7 NUMERICAL EXPERIMENTS

## 7.1 LINEARLY SEPARABLE DATASETS

**Experiments on Synthetic Dataset.** We initiate our experimental evaluation with two synthetic linearly separable datasets, as depicted in Fig. 2. For the two synthetic datasets, the value of $\gamma^\star$ is explicit, and as such, we can explicitly compute the margin gap. To ensure a fair comparison, we maintain the same step size $\eta = 1$ for GD, NGD, and PRGD. Following the guidelines provided in Theorem 6.2, we employ PRGD(exp) with hyperparameters $T_{k+1} - T_k \equiv 5$, $R_k = R_0 \times 1.2^k$. To illustrate the role of the progressive radius, we also examine PRGD(poly) configured with $T_{k+1} - T_k \equiv 5$, $R_k = R_0 \times k^{1.2}$, where the progressive radius increases polynomially. For more experimental details, refer to Appendix E.

The experimental results are provided in Fig. 2. Consistent with Theorem 6.2, PRGD(exp) indeed maximizes the margin exponentially fast, and surprisingly, PRGD(poly) also performs equally well for this task. In contrast, NGD and GD reduce the margin gaps significantly slower, which substantiates our Theorem 6.4.

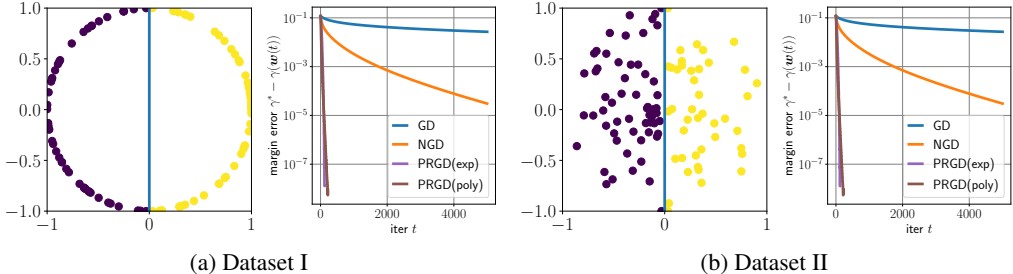

(a) Dataset I          (b) Dataset II

Figure 2: Comparison of margin Maximization rates of different algorithms on Synthetic datasets.

**Experiments on Real-World Datasets.** In this case, we extend our experiments to real-world datasets. Specifically, we employ the `digit` datasets from `Sklearn`, which are image classification tasks with $d = 64$, $n = 300$. In this real-world setting, we lack prior knowledge of the exact $\gamma^\star$. Instead, we approximate $\gamma^\star$ by employing $\gamma(\boldsymbol{w}(t))$ obtained by a sufficiently trained NGD. In real experiments, we test both PRGD(exp) and PRGD(poly) and consistently observe that the latter performs much better. Therefore, in this experiment, we employ a modified variant of PRGD with slower progressive norms: $R_k = R_0 \cdot k^\alpha$, $T_{k+1} - T_k = T_0 \cdot k^\beta$ where $R_0, T_0, \alpha, \beta$ are hyperparameters to be tuned.

The numerical results with well-tuned hyperparameters are presented in Fig. 3. It is evident that, in this real-world dataset, PRGD consistently beats GD and NGD in terms of margin maximization rates.

## 7.2 LINEARLY NON-SEPARABLE DATASETS AND DEEP NEURAL NETWORKS

In this subsection, we further explore the practical performance of PRGD for datasets that are not linearly separable. In the first experiment, we still consider linear models but for classifying a linearly non-separable dataset, `Cancer` in `Sklearn`, and we employ the same PRGD technique as used in real-world linearly separable datasets. For the second experiment, we examine the performance of

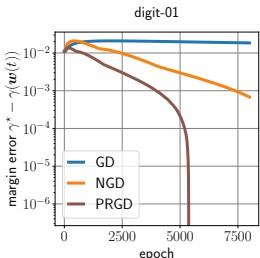 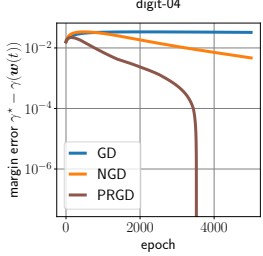

Figure 3: Comparison of margin Maximization rates of different algorithms on `digit` (real-word) datasets. (Left) the results on `digit-01` dataset; (Right) the results on `digit-04` dataset.

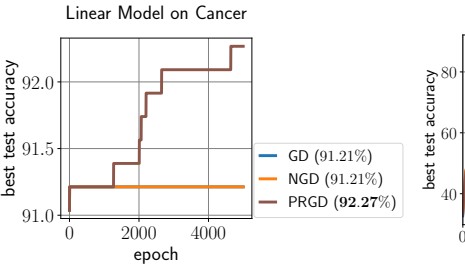 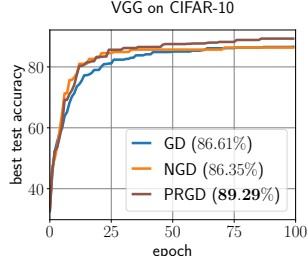

(a) Linear Model on `Cancel`         (b) VGG on Cifar-10

Figure 4: Comparison of the generalization performance of GD, NGD, and PRGD for non-linearly separable datasets and deep neural networks.

PRGD for VGG network (Simonyan & Zisserman, 2015) on the full CIFAR-10 dataset (Krizhevsky & Hinton, 2009), without employing any explicit regularization. Additionally, in this setting, we employ mini-batch stochastic gradient instead of the full gradient for these algorithms, and we also fine-tune the learning rate of GD,NGD, and PRGD. Both of these two algorithms share the same learning rate scheduling strategy as described in Lyu & Li (2019). As for the hyperparameter strategy of PRGD, we still follow the same strategy as used on the real-world linearly separable datasets.

The numerical results are presented in Fig 4a and Fig 4b, respectively. One can see that that our PRGD algorithm outperforms GD and NGD for both tasks.

## 8  CONCLUDING REMARK

In this work, we investigate the mechanisms driving the convergence of gradient-based algorithms towards max-margin solutions. Specifically, we elucidate why GD and NGD can only achieve polynomially fast margin maximization by examining the properties of the velocity field linked to (normalized) gradients. This analysis inspires the design of a novel algorithm called PRGD that significantly accelerates the process of margin maximization. To substantiate our theoretical claims, we offer both synthetic and real-world experimental results, thereby underscoring the potential practical utility of our approach. Looking ahead, an intriguing avenue for future research is the application of progressive norm rescaling techniques to state-of-the-art real-world models. It would be worthwhile to explore how PRGD can synergize with other explicit regularization techniques, such as data augmentation, dropout, and sharpness-aware minimization (Foret et al., 2020).

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

# Appendix

## A  PROOFS IN SECTION 4

**Dataset 2.** The dataset is $\mathcal{S} = \{(\boldsymbol{x}_1, y_1), (\boldsymbol{x}_2, y_2)\}$ where $\boldsymbol{x}_1 = (0, 1)^\top$, $y_1 = 1$, $\boldsymbol{x}_2 = (0, -1)^\top$, and $y_2 = -1$.

**Proposition A.1.** *Consider Dataset 2. Then the max-margin direction is $\boldsymbol{w}^\star = (0, 1)^\top$ with margin $\gamma^\star = 1$. Moreover, the regularization path coincides with the max-margin direction:*

$$\frac{\boldsymbol{w}^\star_{\mathrm{reg}}(B)}{B} \equiv \boldsymbol{w}^\star = (0, 1)^\top, \quad \forall B > 0.$$

**Theorem A.2.** *Consider Dataset 2. Then NGD (3) can only maximize the margin polynomially fast, while PRGD (Alg 1) can maximize the margin exponentially fast. Specifically,*

*(I) Let $\boldsymbol{w}(t)$ be trained by NGD (3) with $\eta = 1$. Then the margin is maximized at **polynomial** rate:*

$$\gamma^\star - \gamma(\boldsymbol{w}(t)) = \Theta\left(1/t^2\right);$$

*(II) Let $\boldsymbol{w}(t)$ be trained by PRGD (Algorithm 1) with $\eta = 1$. If we choose $R_k \equiv R$ and $T_k = \Theta(k)$, then the margin is maximized at **exponential** rate:*

$$\gamma^\star - \gamma(\boldsymbol{w}(t)) = e^{-\Theta(t/R)}.$$

*Proof of Theorem A.2.* Following the Proof of Proposition A.1, we have:

$$-\frac{\nabla\mathcal{L}(\boldsymbol{w})}{\mathcal{L}(\boldsymbol{w})} = \begin{pmatrix} 0 \\ 1 \end{pmatrix}.$$

We assume $w_1(0) \neq 0$. Without loss of generality, we can assume $w_1(0) > 0$ and $w_2(0) > 0$.

Step I. Proof for NGD. For NGD, it holds that:

$$w_1(t+1) = w_1(t),$$
$$w_2(t+1) = w_2(t) + 1.$$

It is easy to verify that $w_1(t) = w_1(0)$ and $w_2(t) = w_2(0) + t$ hold for any $t \geq 0$, which means

$$\frac{w_1(t+1)}{w_2(t+1)} = \frac{w_1(0)}{w_2(0) + t}.$$

From the definition of margin, we have:

$$\gamma(\boldsymbol{w}(t)) - \gamma^* = \left\langle \frac{\boldsymbol{w}(t)}{\|\boldsymbol{w}(t)\|}, \boldsymbol{z}_1 \right\rangle - 1 = \frac{w_2(t)}{\sqrt{w_1^2(t) + w_2^2(t)}} - 1$$

$$= \left(1 + \frac{w_1^2(t)}{w_2^2(t)}\right)^{-1/2} - 1 = -\Theta\left(\frac{w_1^2(t)}{w_2^2(t)}\right) = -\Theta\left(\frac{1}{t^2}\right).$$

**Step II. Proof for PRGD.** For PRGD with fixed $R_k \equiv R > 0$, it is exactly PGD: $\boldsymbol{w}_{t+1} = \mathrm{Proj}_{\mathbb{B}(\mathbf{0},R)}(\boldsymbol{w}(t))$.

Due to $w_1(0) > 0$ and $w_2(0) > 0$, it is easy to verify that $\|\boldsymbol{w}(t)\| = R$ holds for any $t \geq R$. Therefore, for any $t \geq R$, we have:

$$w_1(t+1) = R \frac{w_1(t)}{\sqrt{w_1^2(t) + (w_2(t)+1)^2}},$$

$$w_2(t+1) = R \frac{w_2(t)+1}{\sqrt{w_1^2(t) + (w_2(t)+1)^2}}.$$

These dynamics imply that for any $t \geq R$,

$$\frac{w_2(t+1)}{w_1(t+1)} = \frac{w_2(t)}{w_1(t)} + \frac{1}{w_1(t)} = \frac{w_2(t)}{w_1(t)} + \frac{\sqrt{w_1^2(t)+w_2^2(t)}}{Rw_1(t)} = \frac{w_2(t)}{w_1(t)} + \frac{1}{R}\sqrt{1 + \frac{w_2^2(t)}{w_1^2(t)}}.$$

On the one hand, we derive the lower bound for $w_2(t)/w_1(t)$:

$$\frac{w_2(t+1)}{w_1(t+1)} \geq \frac{w_2(t)}{w_1(t)} + \frac{1}{R}\frac{w_2(t)}{w_1(t)} = \left(1 + \frac{1}{R}\right)\frac{w_2(t)}{w_1(t)}, \ \forall t \geq R.$$

which means

$$\frac{w_2(t)}{w_1(t)} \geq \left(1 + \frac{1}{R}\right)^{t-\lceil R\rceil} \frac{w_2(\lceil R\rceil)}{w_1(\lceil R\rceil)}, \ \forall t \geq \lceil R\rceil.$$

On the other hand, we derive the upper bound for $w_2(t)/w_1(t)$. Notably, $\frac{w_2(t+1)}{w_1(t+1)} \geq \frac{w_2(t)}{w_1(t)} + \frac{1}{R}$ holds for any $t \geq R$. Thus, for any $t \geq 2\lceil R\rceil$,

$$\frac{w_2(t)}{w_1(t)} \geq \frac{w_2(2\lceil R\rceil)}{w_1(2\lceil R\rceil)} \geq \frac{w_2(2\lceil R\rceil - 1)}{w_1(2\lceil R\rceil - 1)} + \frac{1}{R} \geq \cdots \geq \frac{w_2(\lceil R\rceil)}{w_1(\lceil R\rceil)} + \frac{\lceil R\rceil}{R} \geq 1.$$

Hence, we have the following upper bound for any $t \geq 2\lceil R\rceil$:

$$\frac{w_2(t+1)}{w_1(t+1)} \leq \frac{w_2(t)}{w_1(t)} + \frac{1}{R}\sqrt{\frac{w_2^2(t)}{w_1^2(t)} + \frac{w_2^2(t)}{w_1^2(t)}} = \left(1 + \frac{\sqrt{2}}{R}\right)\frac{w_2(t)}{w_1(t)},$$

which means

$$\frac{w_2(t)}{w_1(t)} \leq \left(1 + \frac{\sqrt{2}}{R}\right)^{t-2\lceil R\rceil} \frac{w_2(2\lceil R\rceil)}{w_1(2\lceil R\rceil)}, \ \forall t \geq 2\lceil R\rceil.$$

Combining the lower bound and upper bound, we obtain the tight estimate:

$$\frac{w_2(t)}{w_1(t)} = \exp\left(\Theta(t)\log\left(1 + \frac{1}{\Theta(R)}\right)\right) = e^{\Theta(t/R)}.$$

From the definition of margin, we have:

$$\gamma(\boldsymbol{w}(t)) - \gamma^* = \left\langle \frac{\boldsymbol{w}(t)}{\|\boldsymbol{w}(t)\|}, \boldsymbol{z}_1 \right\rangle - 1 = \frac{w_2(t)}{\sqrt{w_1^2(t) + w_2^2(t)}} - 1$$

$$= \left(1 + \frac{w_1^2(t)}{w_2^2(t)}\right)^{-1/2} - 1 = -\Theta\left(\frac{w_1^2(t)}{w_2^2(t)}\right) = -e^{-\Theta(t/R)}.$$

$\square$

*Proof of Proposition 4.1.*
For simplicity, we denote $z_1 = x_1 y_1$ and $z_2 = x_2 y_2$.

$$\mathcal{L}(w) = \frac{1}{3}\left(e^{-w^\top z_1} + 2e^{-w^\top z_2}\right) = \frac{1}{3}e^{-w_2\gamma}\left(e^{-w_1\sqrt{1-\gamma^2}} + 2e^{w_1\sqrt{1-\gamma^2}}\right).$$

$$\nabla\mathcal{L}(w) = \begin{pmatrix} \frac{1}{3}e^{-w_2\gamma}\sqrt{1-\gamma^2}\left(-e^{-w_1\sqrt{1-\gamma^2}} + 2e^{w_1\sqrt{1-\gamma^2}}\right) \\ -\frac{1}{3}e^{-w_2\gamma}\gamma\left(e^{-w_1\sqrt{1-\gamma^2}} + 2e^{w_1\sqrt{1-\gamma^2}}\right) \end{pmatrix}.$$

For any fixed $R > 0$, we will calculate the regularized solution in the ball $\|w\|_2 \leq R$.

From the expression of $\nabla\mathcal{L}(w)$, we know $\nabla\mathcal{L}(w) \neq 0$ for any $w \in \mathbb{R}^d$. Hence, it must holds $\|w_{\text{reg}}^*(R)\|_2 = R$. Moreover, we can determine the signal of $w_{\text{reg},1}^*(R)$ and $w_{\text{reg},2}^*(R)$. From the symmetry of the $\ell_2$ ball, we know $w_{\text{reg},1}^*(R) < 0$ and $w_{\text{reg},2}^*(R) > 0$. This is because: if $w_{\text{reg},1}^*(R) > 0$, then $\mathcal{L}(-w_{\text{reg},1}^*(R), w_{\text{reg},2}^*(R)) < \mathcal{L}(w_{\text{reg},1}^*(R), w_{\text{reg},2}^*(R))$, which is contradict to the optimum of $w_{\text{reg}}^*(R)$.

Then from the optimum and differentiability, we have

$$\frac{\langle w_{\text{reg}}^*(R), -\nabla\mathcal{L}\left(w_{\text{reg}}^*(R)\right)\rangle}{R\left\|\nabla\mathcal{L}\left(w_{\text{reg}}^*(R)\right)\right\|_2} = 1,$$

which means

$$w_{\text{reg}}^*(R) \mathbin{/\mkern-5mu/} \nabla\mathcal{L}\left(w_{\text{reg}}^*(R)\right), \quad \langle w_{\text{reg}}^*(R), \nabla\mathcal{L}\left(w_{\text{reg}}^*(R)\right)\rangle < 0.$$

For simplicity, we use the notation $w_1(R) := w_{\text{reg},1}^*(R)$, $w_2(R) := w_{\text{reg},2}^*(R)$ in the proof below.

By a straightforward calculation and taking the square, we have

$$\frac{(1-\gamma^2)\left(e^{-2w_1(B)\sqrt{1-\gamma^2}} + 4e^{2w_1(B)\sqrt{1-\gamma^2}} - 4\right)}{\gamma^2\left(e^{-2w_1(R)\sqrt{1-\gamma^2}} + 4e^{2w_1\sqrt{1-\gamma^2}} + 4\right)} = \frac{w_1^2(B)}{w_2^2(R)} = \frac{w_1^2(R)}{R^2 - w_1^2(R)},$$

which is equivalent to

$$\frac{R^2}{w_1^2(R)} = \frac{1}{1-\gamma^2} + \frac{8\gamma^2}{(1-\gamma^2)\left(e^{-2w_1(R)\sqrt{1-\gamma^2}} + 4e^{2w_1(R)\sqrt{1-\gamma^2}} - 4\right)}. \tag{4}$$

With the help of Lemma A.3, we know

$$\lim_{R\to\infty}\left\langle w^*, \frac{w_{\text{reg}}^*(R)}{R}\right\rangle = \lim_{R\to\infty}\frac{w_2(R)}{\sqrt{w_1^2(R) + w_2^2(R)}} = 1,$$

which means $\lim_{R\to\infty}\frac{w_1^2(R)}{R^2} = 0$. Then taking $R \to \infty$ in (4), we have

$$\lim_{R\to\infty}\left(e^{-2w_1(R)\sqrt{1-\gamma^2}} + 4e^{2w_1(R)\sqrt{1-\gamma^2}}\right) = 4.$$

A straight-forward calculation gives us

$$\lim_{R\to\infty} w_1(R) = -\frac{\log 2}{2\sqrt{1-\gamma^2}}.$$

Following the proof, we have

$$-\frac{\nabla\mathcal{L}(w)}{\mathcal{L}(w)} = \begin{pmatrix} \sqrt{1-\gamma^2}\left(1 - 2e^{2w_1\sqrt{1-\gamma^2}}\right) / \left(1 + 2e^{2w_1\sqrt{1-\gamma^2}}\right) \\ \gamma \end{pmatrix}.$$

For simplicity, we assume $w(0) = 0$. For other cases, the proof is similar (Ji & Telgarsky, 2021).

Step I. Proof for NGD. For NGD, it holds that:

$$w_1(t+1) = w_1(t) + \sqrt{1-\gamma^2}\left(1 - 2e^{2w_1(t)\sqrt{1-\gamma^2}}\right)\Big/\left(1 + 2e^{2w_1(t)\sqrt{1-\gamma^2}}\right),$$

$$w_2(t+1) = w_2(t) + \gamma.$$

It is worth noticing that the dynamics of $w_1(t)$ and $w_2(t)$ are decoupled. For $w_2(t)$, it is easy to verify that $w_2(t) = \gamma t, \forall t \geq 1$. As for $w_1(t)$, we will estimate the uniform upper and lower bounds.

For simplicity, we denote $x(t) := 2w_1(t)\sqrt{1-\gamma^2} + \log 2$. From the dynamics of $w_1(t)$, the dynamics of $x(t)$ are

$$x(t+1) = x(t) + 2(1-\gamma^2)\frac{1-e^{x(t)}}{1+e^{x(t)}} = x(t) + 2(1-\gamma^2)\left(\frac{2}{1+e^{x(t)}} - 1\right).$$

Then we will prove that $|x(t)| \leq \frac{1}{2}\log 2$ holds for $t \geq 1$ by induction.

From $x(0) = \log 2$, we have $x(1) = \log 2 - \frac{2(1-\gamma^2)}{3} \in \left[-\frac{1}{2}\log 2, \frac{1}{2}\log 2\right]$.

Assume that $x(t) \in \left[-\frac{1}{2}\log 2, \frac{1}{2}\log 2\right]$ holds for any $t \leq k$, and we denote $h(x) := x + 2(1 - \gamma^2)\left(\frac{2}{1+e^x} - 1\right)$. Then with the help of Lemma D.2, the following estimate holds for $t = k+1$:

$$x(k+1) = h(x(k)) \leq h\left(\frac{1}{2}\log 2\right) = \frac{1}{2}\log 2 + 2(1-\gamma^2)\frac{1-\sqrt{2}}{1+\sqrt{2}} < \frac{1}{2}\log 2;$$

$$x(k+1) = h(x(k)) \geq h\left(-\frac{1}{2}\log 2\right) = -\frac{1}{2}\log 2 + 2(1-\gamma^2)\frac{\sqrt{2}-1}{\sqrt{2}+1} > -\frac{1}{2}\log 2.$$

By induction, we have proved that $x(t) \in \left[-\frac{1}{2}\log 2, \frac{1}{2}\log 2\right]$ holds for any $t \geq 1$. This implies that $w_1(t) \in \left[-\frac{3\log 2}{4\sqrt{1-\gamma^2}}, -\frac{\log 2}{4\sqrt{1-\gamma^2}}\right]$ holds for any $t \geq 1$. Hence,

$$-\frac{3\log 2}{4t\gamma\sqrt{1-\gamma^2}} \leq \frac{w_1(t)}{w_2(t)} \leq -\frac{\log 2}{4t\gamma\sqrt{1-\gamma^2}}, \quad \forall t \geq 1.$$

From the definition of directional convergence, we have:

$$\left\|\frac{\boldsymbol{w}(t)}{\|\boldsymbol{w}(t)\|} - \boldsymbol{w}^\star\right\| = \sqrt{2\left(1 - \left\langle\frac{\boldsymbol{w}(t)}{\|\boldsymbol{w}(t)\|}, \boldsymbol{e}_2\right\rangle\right)} = \sqrt{2\left(1 - \frac{w_2(t)}{\sqrt{w_1^2(t)+w_2^2(t)}}\right)}$$

$$= \sqrt{2\left(1 - \frac{1}{\sqrt{\frac{w_1^2(t)}{w_2^2(t)}+1}}\right)} = \Theta\left(\left|\frac{w_1(t)}{w_2(t)}\right|\right) = \Theta\left(\frac{1}{t}\right).$$

From the definition of margin, we have:

$$\gamma(\boldsymbol{w}(t)) - \gamma^* = \min_{i\in[2]}\left\langle\frac{\boldsymbol{w}(t)}{\|\boldsymbol{w}(t)\|}, \boldsymbol{z}_i\right\rangle - \gamma = \left\langle\frac{\boldsymbol{w}(t)}{\|\boldsymbol{w}(t)\|}, \boldsymbol{z}_1\right\rangle - \gamma$$

$$= \frac{w_1(t)\sqrt{1-\gamma^2} + w_2(t)\gamma}{\sqrt{w_1^2(t)+w_2^2(t)}} - \gamma = \frac{\frac{w_1(t)}{w_2(t)}\sqrt{1-\gamma^2} + \gamma}{\sqrt{\frac{w_1^2(t)}{w_2^2(t)}+1}} - \gamma$$

$$= \frac{\frac{w_1(t)}{w_2(t)}\sqrt{1-\gamma^2}}{\sqrt{\frac{w_1^2(t)}{w_2^2(t)}+1}} + \gamma\left(\left(\frac{w_1^2(t)}{w_2^2(t)}+1\right)^{-1/2} - 1\right) = \Theta\left(\frac{w_1(t)}{w_2(t)}\right) - \Theta\left(\frac{w_1^2(t)}{w_2^2(t)}\right)$$

$$= \Theta\left(\frac{w_1(t)}{w_2(t)}\right) = -\Theta\left(\frac{1}{t}\right).$$

Step II. Proof for PRGD. For PRGD, to maximize margin exponentially fast, we only need to select $R_k = e^{\Theta(k)}$ and $T_k = \Theta(k)$. Notice that the choices of $R_k$ and $T_k$ are not unique. For simplicity, we use the following choice to make our proof clear.

- Phase I. We run NGD for $t < T^{\mathrm{I}} = \lceil 1/\gamma \rceil$. (It is worth noting that this phase is exactly to run PRGD for $2T_1$ steps, with $R_k \equiv T^{\mathrm{I}} + 1$ and $T_{k+1} = T_k + 2$.)

- Phase II. We run PRGD for $t \geq T^{\mathrm{I}}$. For a fixed integer $D \geq 10$, we select $T_k$ and $R_k$ such that:

$$T_0 = T^{\mathrm{I}}; \quad T_{k+1} = T_k + D, \ \forall k \geq 0; \quad R_k = \frac{D \, \|\boldsymbol{w}(T_k)\|}{|w_1(T_k)|}, \ \forall k \geq 0.$$

Recalling our proof in Step I, at the end of Phase I, it holds that $w_1(T^{\mathrm{I}}) \in \left[ -\frac{3 \log 2}{4\sqrt{1-\gamma^2}}, -\frac{\log 2}{4\sqrt{1-\gamma^2}} \right]$ and $w_2(T^{\mathrm{I}}) = T^{\mathrm{I}}$. Then we analyze Phase II. We will prove the following statements by induction.

- (S1). $\|\boldsymbol{w}(t)\| = R_k$ holds for any $T_k \leq t < T_{k+1}$;

- (S2). $w_1(T_k) = -D$ holds for any $k \geq 0$;

- (S3). $\leq w_1(t) \leq$ holds for any $T_k \leq t < T_{k+1}$;

- (S4). $\leq \frac{w_1(T_k)}{w_2(T_k)} \leq$ holds for any $k \geq 0$;

- (S5). $\frac{w_1(T_k)}{w_2(T_k)} \leq \frac{w_1(t)}{w_2(t)} \leq \frac{w_1(s)}{w_2(s)} \leq \frac{w_1(T_{k+1})}{w_2(T_{k+1})}$ for any $T_k \leq t \leq s \leq T_{k+1}$

First, according to PRGD's update, for $k = 0$ ($T_0 = T^{\mathrm{I}}$), we have

$\boldsymbol{w}(T_0 + 1) = R_0 \frac{\boldsymbol{w}(T_0)}{\|\boldsymbol{w}(T_0)\|} = \frac{D}{|w_1(T_0)|} \boldsymbol{w}(T_0)$

Assume (S1)~(S5) hold for any $T_k (k \leq K - 1)$ and any $t < T_K$. Then we need to prove that (S1)~(S5) hold for $T_K$ and $t \in [T_K, T_{K+1})$.

Hence, we have proved (S1)(S2)(S3). Consequently, these statements imply that:

$$\frac{w_1(t)}{w_2(t)} = -e^{-\Theta(t)}$$

From the definition of directional convergence, we have:

$$\left\| \frac{\boldsymbol{w}(t)}{\|\boldsymbol{w}(t)\|} - \boldsymbol{w}^\star \right\| = \sqrt{2\left(1 - \left\langle \frac{\boldsymbol{w}(t)}{\|\boldsymbol{w}(t)\|}, \boldsymbol{e}_2 \right\rangle\right)} = \sqrt{2\left(1 - \frac{w_2(t)}{\sqrt{w_1^2(t) + w_2^2(t)}}\right)}$$

$$= \sqrt{2\left(1 - \frac{1}{\sqrt{\frac{w_1^2(t)}{w_2^2(t)} + 1}}\right)} = \Theta\left(\left|\frac{w_1(t)}{w_2(t)}\right|\right) = e^{-\Theta(t)}.$$

From the definition of margin, we have:

$$\gamma(\boldsymbol{w}(t)) - \gamma^* = \min_{i \in [2]} \left\langle \frac{\boldsymbol{w}(t)}{\|\boldsymbol{w}(t)\|}, \boldsymbol{z}_i \right\rangle - \gamma = \left\langle \frac{\boldsymbol{w}(t)}{\|\boldsymbol{w}(t)\|}, \boldsymbol{z}_1 \right\rangle - \gamma$$

$$= \frac{w_1(t)\sqrt{1-\gamma^2} + w_2(t)\gamma}{\sqrt{w_1^2(t) + w_2^2(t)}} - \gamma = \frac{\frac{w_1(t)}{w_2(t)}\sqrt{1-\gamma^2} + \gamma}{\sqrt{\frac{w_1^2(t)}{w_2^2(t)} + 1}} - \gamma$$

$$= \frac{\frac{w_1(t)}{w_2(t)}\sqrt{1-\gamma^2}}{\sqrt{\frac{w_1^2(t)}{w_2^2(t)} + 1}} + \gamma\left(\left(\frac{w_1^2(t)}{w_2^2(t)} + 1\right)^{-1/2} - 1\right) = \Theta\left(\frac{w_1(t)}{w_2(t)}\right) - \Theta\left(\frac{w_1^2(t)}{w_2^2(t)}\right)$$

$$= \Theta\left(\frac{w_1(t)}{w_2(t)}\right) = -e^{-\Theta(t)}.$$

$\square$

**Lemma A.3** (Integration of ([Soudry et al., 2018](#); [Ji et al., 2020](#))). *For problem* (1)*, Gradient Flow convergences to the $\ell_2$ max-margin direction $\boldsymbol{w}^*$, hence the regularization path also convergences to the $\ell_2$ max-margin solution:* $\lim\limits_{B \to \infty} \frac{\boldsymbol{w}^*_{\mathrm{reg}}(B)}{B} = \boldsymbol{w}^*$.

**Lemma A.4** (Margin error and Directional error). *Under Assumption 3.1, for any $\boldsymbol{w} \in \mathbb{R}^d$, it holds that* $\gamma^\star - \gamma(\boldsymbol{w}) \le \left\| \frac{\boldsymbol{w}}{\|\boldsymbol{w}\|} - \boldsymbol{w}^\star \right\|$.

*Proof of Lemma A.4.*

Let $\boldsymbol{w} \in \mathbb{R}^d$ and denote $i_0 \in \arg\min\limits_{i \in [n]} y_i \left\langle \frac{\boldsymbol{w}}{\|\boldsymbol{w}\|}, \boldsymbol{x}_i \right\rangle$. Then we have:

$$
\begin{aligned}
\gamma^\star - \gamma(\boldsymbol{w}) &= \min_i y_i \langle \boldsymbol{w}^\star, \boldsymbol{x}_i \rangle - \min_i y_i \left\langle \frac{\boldsymbol{w}}{\|\boldsymbol{w}\|}, \boldsymbol{x}_i \right\rangle \\
&= \min_i y_i \langle \boldsymbol{w}^\star, \boldsymbol{x}_i \rangle - y_{i_0} \left\langle \frac{\boldsymbol{w}}{\|\boldsymbol{w}\|}, \boldsymbol{x}_{i_0} \right\rangle \\
&\le y_{i_0} \langle \boldsymbol{w}^\star, \boldsymbol{x}_{i_0} \rangle - y_{i_0} \left\langle \frac{\boldsymbol{w}}{\|\boldsymbol{w}\|}, \boldsymbol{x}_{i_0} \right\rangle = y_{i_0} \left\langle \boldsymbol{w}^\star - \frac{\boldsymbol{w}}{\|\boldsymbol{w}(t)\|}, \boldsymbol{x}_{i_0} \right\rangle \\
&\le \left\| \frac{\boldsymbol{w}}{\|\boldsymbol{w}\|} - \boldsymbol{w}^\star \right\|.
\end{aligned}
$$

$\square$

## B  PROOFS IN SECTION 5

**Lemma B.1.** *Let $H = \frac{1}{\gamma^\star_{\mathrm{sub}} - \gamma^\star} \log\left( \frac{n - |\mathcal{I}|}{|\mathcal{I}|} \right)$. Then for any $h \ge H$,*

$$
|\mathcal{I}| \exp(-h\gamma^\star) \le \sum_{i=1}^n \exp(-h \langle \boldsymbol{w}^\star, y_i \boldsymbol{x}_i \rangle) \le 2|\mathcal{I}| \exp(-h\gamma^\star).
$$

*Proof of Lemma B.1.*

First, notice that $\langle \boldsymbol{w}^\star, y_i \boldsymbol{x}_i \rangle = \gamma^\star$ for any $i \in \mathcal{I}$ and $\langle \boldsymbol{w}^\star, y_j \boldsymbol{x}_j \rangle \ge \gamma^\star_{\mathrm{sub}} > \gamma^\star$ any $j \notin \mathcal{I}$. Therefore, for any

$$
h \ge H = \frac{1}{\gamma^\star_{\mathrm{sub}} - \gamma^\star} \log\left( \frac{n - |\mathcal{I}|}{|\mathcal{I}|} \right),
$$

any $i \in \mathcal{I}$ and $j \notin \mathcal{I}$, we have

$$
\frac{\exp(-h \langle \boldsymbol{w}^\star, y_i \boldsymbol{x}_i \rangle)}{\exp(-h \langle \boldsymbol{w}^\star, y_j \boldsymbol{x}_j \rangle)} \ge \frac{\exp(-h\gamma^\star)}{\exp(-h\gamma^\star_{\mathrm{sub}})} = \exp(h(\gamma^\star_{\mathrm{sub}} - \gamma^\star)) \ge \frac{n - |\mathcal{I}|}{|\mathcal{I}|},
$$

which implies that

$$
\begin{aligned}
\sum_{i=1}^n \exp(-h \langle \boldsymbol{w}^\star, y_i \boldsymbol{x}_i \rangle) &= \sum_{i \in \mathcal{I}} \exp(-h \langle \boldsymbol{w}^\star, y_i \boldsymbol{x}_i \rangle) + \sum_{j \notin \mathcal{I}} \exp(-h \langle \boldsymbol{w}^\star, y_j \boldsymbol{x}_j \rangle) \\
&\le |\mathcal{I}| \exp(-h\gamma^\star) + \sum_{j \notin \mathcal{I}} \exp(-h\gamma^\star_{\mathrm{sub}}) \\
&\le |\mathcal{I}| \exp(-h\gamma^\star) + \sum_{j \notin \mathcal{I}} \frac{|\mathcal{I}|}{n - |\mathcal{I}|} \exp(-h\gamma^\star) \\
&= |\mathcal{I}| \exp(-h\gamma^\star) + |\mathcal{I}| \exp(-h\gamma^\star) = 2|\mathcal{I}| \exp(-h\gamma^\star).
\end{aligned}
$$

As for the left inequality, we only notice

$$
\sum_{i=1}^n \exp(-h \langle \boldsymbol{w}^\star, y_i \boldsymbol{x}_i \rangle) \ge \sum_{i \in \mathcal{I}} \exp(-h \langle \boldsymbol{w}^\star, y_i \boldsymbol{x}_i \rangle) = |\mathcal{I}| \exp(-h\gamma^\star).
$$

Hence, we complete the proof.

$\square$

*Proof of Theorem 5.4.*
Without loss of generality, we can assume $\mathrm{span}\{\boldsymbol{x}_1, \cdots, \boldsymbol{x}_n\} = \mathbb{R}^d$. This is because: GD, NGD, and PRGD can only evaluate in $\mathrm{span}\{\boldsymbol{x}_i : i \in [n]\}$, i.e. $\boldsymbol{w}(t) \in \mathrm{span}\{\boldsymbol{x}_i : i \in [n]\}$. If $\mathrm{span}\{\boldsymbol{x}_1, \cdots, \boldsymbol{x}_n\} \neq \mathbb{R}^d$, we only need to change the proof in the subspace $\mathrm{span}\{\boldsymbol{x}_1, \cdots, \boldsymbol{x}_n\}$.

Therefore, from the definition of $\mathbb{C}(D; H)$, it holds that

$$\mathbb{C}(D; H) = \left\{ h\boldsymbol{w}^\star + D\boldsymbol{v} : h \geq H, \boldsymbol{v} \in \mathbb{S}^{d-1}, \boldsymbol{v} \perp \boldsymbol{w}^\star \right\}.$$

Step I. Strip out the important ingredients.

First, following Lemma B.1, we select $H = \frac{1}{\gamma_{\mathrm{sub}}^\star - \gamma^\star} \log\left(\frac{n - |\mathcal{I}|}{|\mathcal{I}|}\right)$.

Then for any $\boldsymbol{w} \in \mathbb{C}(D; H)$, we have:

$$
\begin{aligned}
\left\langle \frac{\nabla\mathcal{L}(\boldsymbol{w})}{\mathcal{L}(\boldsymbol{w})}, \frac{\mathcal{P}_\perp(\boldsymbol{w})}{\|\mathcal{P}_\perp(\boldsymbol{w})\|} \right\rangle &= \left\langle \frac{\nabla\mathcal{L}(\boldsymbol{w})}{\mathcal{L}(\boldsymbol{w})}, \boldsymbol{v} \right\rangle \\
&= \left\langle \frac{\frac{1}{n}\sum_{i=1}^n (-y_i\boldsymbol{x}_i)\exp(-\langle\boldsymbol{w}, y_i\boldsymbol{x}_i\rangle)}{\frac{1}{n}\sum_{i=1}^n \exp(-y_i\langle\boldsymbol{w}, \boldsymbol{x}_i\rangle)}, \boldsymbol{v} \right\rangle \\
&= \frac{\frac{1}{n}\sum_{i=1}^n \langle\boldsymbol{v}, -y_i\boldsymbol{x}_i\rangle \exp\left(-h\langle\boldsymbol{w}^\star, y_i\boldsymbol{x}_i\rangle\right)\exp\left(-D\langle\boldsymbol{v}, y_i\boldsymbol{x}_i\rangle\right)}{\frac{1}{n}\sum_{i=1}^n \exp\left(-h\langle\boldsymbol{w}^\star, y_i\boldsymbol{x}_i\rangle\right)\exp\left(-D\langle\boldsymbol{v}, y_i\boldsymbol{x}_i\rangle\right)} \\
&\geq \frac{\sum_{i=1}^n \langle\boldsymbol{v}, -y_i\boldsymbol{x}_i\rangle \exp\left(-h\langle\boldsymbol{w}^\star, y_i\boldsymbol{x}_i\rangle\right)\exp\left(-D\langle\boldsymbol{v}, y_i\boldsymbol{x}_i\rangle\right)}{\sum_{i=1}^n \exp\left(-h\langle\boldsymbol{w}^\star, y_i\boldsymbol{x}_i\rangle\right)\exp\left(D\right)} \\
&\overset{\text{Lemma B.1}}{\geq} \frac{\sum_{i=1}^n \langle\boldsymbol{v}, -y_i\boldsymbol{x}_i\rangle \exp\left(-h\langle\boldsymbol{w}^\star, y_i\boldsymbol{x}_i\rangle\right)\exp\left(-D\langle\boldsymbol{v}, y_i\boldsymbol{x}_i\rangle\right)}{|\mathcal{I}|\exp\left(-h\gamma^\star\right)\exp(D)} \\
&\geq \frac{\sum_{i\in\mathcal{I}} \langle\boldsymbol{v}, -y_i\boldsymbol{x}_i\rangle \exp\left(-h\gamma^\star\right)\exp\left(-D\langle\boldsymbol{v}, y_i\boldsymbol{x}_i\rangle\right)}{|\mathcal{I}|\exp\left(-h\gamma^\star\right)\exp(D)} \\
&= \frac{1}{|\mathcal{I}|\exp(D)} \sum_{i\in\mathcal{I}} \langle\boldsymbol{v}, -y_i\boldsymbol{x}_i\rangle \exp\left(-D\langle\boldsymbol{v}, y_i\boldsymbol{x}_i\rangle\right).
\end{aligned}
$$

Thus, we only need to derive the lower bound of $\sum_{i\in\mathcal{I}} \langle\boldsymbol{v}, -y_i\boldsymbol{x}_i\rangle \exp\left(-D\langle\boldsymbol{v}, y_i\boldsymbol{x}_i\rangle\right)$ for any $\{\boldsymbol{v} \in \mathbb{S}^{d-1} : \boldsymbol{v} \perp \boldsymbol{w}^\star\}$.

Step II. Uniform Lower bound of $\sum_{i\in\mathcal{I}} \langle\boldsymbol{v}, -y_i\boldsymbol{x}_i\rangle \exp\left(-D\langle\boldsymbol{v}, y_i\boldsymbol{x}_i\rangle\right)$.

First, recalling Assumption 5.4 (ii) and the remark, there exist $\alpha_i > 0$ $(i \in \mathcal{I})$ such that $\boldsymbol{w}^\star = \sum_{i\in\mathcal{I}} \alpha_i y_i \boldsymbol{x}_i$, where $\sum_{i\in\mathcal{I}} \alpha_i = 1$. Therefore, we have

$$\sum_{i\in\mathcal{I}} \langle\boldsymbol{v}, -y_i\boldsymbol{x}_i\rangle \exp\left(-D\langle\boldsymbol{v}, y_i\boldsymbol{x}_i\rangle\right) > \sum_{i\in\mathcal{I}} \alpha_i \langle\boldsymbol{v}, -y_i\boldsymbol{x}_i\rangle \exp\left(-D\langle\boldsymbol{v}, y_i\boldsymbol{x}_i\rangle\right).$$

For simplicity, we denote the function $\phi(z) := ze^{Dz}$. Then

$$\sum_{i\in\mathcal{I}} \langle\boldsymbol{v}, -y_i\boldsymbol{x}_i\rangle \exp\left(-D\langle\boldsymbol{v}, y_i\boldsymbol{x}_i\rangle\right) = \sum_{i\in\mathcal{I}} \alpha_i \phi\left(-\langle\boldsymbol{v}, y_i\boldsymbol{x}_i\rangle\right).$$

Now we select $D = 1$. With the help of Lemma D.3, $\phi(\cdot)$ is $e^{-1}$-strongly convex in $z \in [-1, 1]$, which means that

$$\phi(z_1) \geq \phi(z_2) + \phi'(z_2)(z_1 - z_2) + \frac{1}{2e}(z_1 - z_2)^2, \ \forall z_1, z_2 \in [-1, 1].$$

Therefore, for each $i \in \mathcal{I}$, we have

$$\phi\left(\langle\boldsymbol{v}, -y_i\boldsymbol{x}_i\rangle\right) \geq \phi(0) + \phi'(0)\langle\boldsymbol{v}, -y_i\boldsymbol{x}_i\rangle + \frac{1}{2e}\langle\boldsymbol{v}, -y_i\boldsymbol{x}_i\rangle^2$$

$$= \langle \boldsymbol{v}, -y_i \boldsymbol{x}_i \rangle + \frac{1}{2e} \langle \boldsymbol{v}, -y_i \boldsymbol{x}_i \rangle^2.$$

Taking the $\alpha_i$-weighted sum over $i \in \mathcal{I}$ and noticing $\langle \boldsymbol{v}, \boldsymbol{w}^\star \rangle = 0$, we have

$$\sum_{i \in \mathcal{I}} \alpha_i \phi\left(\langle \boldsymbol{v}, -y_i \boldsymbol{x}_i \rangle\right) \geq \sum_{i \in \mathcal{I}} \alpha_i \langle \boldsymbol{v}, -y_i \boldsymbol{x}_i \rangle + \sum_{i \in \mathcal{I}} \frac{\alpha_i}{2e} \langle \boldsymbol{v}, -y_i \boldsymbol{x}_i \rangle^2$$

$$= \langle \boldsymbol{v}, \boldsymbol{w}^\star \rangle + \sum_{i \in \mathcal{I}} \frac{\alpha_i}{2e} \langle \boldsymbol{v}, -y_i \boldsymbol{x}_i \rangle^2 = \sum_{i \in \mathcal{I}} \frac{\alpha_i}{2e} \langle \boldsymbol{v}, -y_i \boldsymbol{x}_i \rangle^2$$

$$\geq \frac{1}{2e} \left(\min_{i \in \mathcal{I}} \alpha_i\right) \left(\boldsymbol{v}^\top \sum_{i \in \mathcal{I}} \left(\boldsymbol{x}_i \boldsymbol{x}_i^\top\right) \boldsymbol{v}\right) \geq \frac{\left(\min\limits_{i \in \mathcal{I}} \alpha_i\right) \lambda_{\min}\left(\sum\limits_{i \in \mathcal{I}} \boldsymbol{x}_i \boldsymbol{x}_i^\top\right)}{2e}.$$

Recalling Assumption 5.4 (i), it holds $\mathrm{rank}\{\boldsymbol{x}_i : i \in \mathcal{I}\} = \mathrm{rank}\{\boldsymbol{x}_i : i \in [n]\} = d$, which implies $\lambda_{\min}\left(\sum\limits_{i \in \mathcal{I}} \boldsymbol{x}_i \boldsymbol{x}_i^\top\right) > 0$. Hence, we obtain the uniform lower bound:

$$\inf_{\boldsymbol{v} \in \{\boldsymbol{v} \in \mathbb{S}^{d-1} : \boldsymbol{v} \perp \boldsymbol{w}^\star\}} \sum_{i \in \mathcal{I}} \langle \boldsymbol{v}, -y_i \boldsymbol{x}_i \rangle \exp\left(-D \langle \boldsymbol{v}, y_i \boldsymbol{x}_i \rangle\right) \geq \frac{\left(\min\limits_{i \in \mathcal{I}} \alpha_i\right) \lambda_{\min}\left(\sum\limits_{i \in \mathcal{I}} \boldsymbol{x}_i \boldsymbol{x}_i^\top\right)}{2e} > 0.$$

**Step III. The final bound.**

We select $D = 1$ and $H = \max\left\{\frac{1}{\gamma_{\mathrm{sub}}^\star - \gamma^\star} \log\left(\frac{n - |\mathcal{I}|}{|\mathcal{I}|}\right), 0\right\}$. Combing our results in Step I and II, for any $\boldsymbol{w} \in \mathbb{C}(D; H)$, it holds that

$$\left\langle \frac{\nabla \mathcal{L}(\boldsymbol{w})}{\mathcal{L}(\boldsymbol{w})}, \frac{\mathcal{P}_\perp(\boldsymbol{w})}{\|\mathcal{P}_\perp(\boldsymbol{w})\|} \right\rangle$$

$$\geq \frac{1}{|\mathcal{I}| e \sum_{i \in \mathcal{I}} \langle \boldsymbol{v}, -y_i \boldsymbol{x}_i \rangle \exp\left(-\langle \boldsymbol{v}, y_i \boldsymbol{x}_i \rangle\right)}$$

$$\geq \frac{\left(\min\limits_{i \in \mathcal{I}} \alpha_i\right) \lambda_{\min}\left(\sum\limits_{i \in \mathcal{I}} \boldsymbol{x}_i \boldsymbol{x}_i^\top\right)}{2e^2 |\mathcal{I}|} > 0.$$

$\square$

## C  PROOFS IN SECTION 6

**Lemma C.1.** *Under Assumption 3.1, it holds that*

$$\gamma^\star \leq \left\langle -\frac{\nabla \mathcal{L}(\boldsymbol{w})}{\mathcal{L}(\boldsymbol{w})}, \boldsymbol{w}^\star \right\rangle \leq 1, \quad \gamma^\star \leq \left\| \frac{\nabla \mathcal{L}(\boldsymbol{w})}{\mathcal{L}(\boldsymbol{w})} \right\| \leq 1, \quad \forall \boldsymbol{w} \in \mathbb{R}^d.$$

*Proof of Lemma C.1.* For any $\boldsymbol{w} \in \mathbb{R}^d$, we have:

$$\left\langle -\frac{\nabla \mathcal{L}(\boldsymbol{w})}{\mathcal{L}(\boldsymbol{w})}, \boldsymbol{w}^\star \right\rangle = \frac{\frac{1}{n} \sum_{i=1}^n e^{-y_i \langle \boldsymbol{w}, \boldsymbol{x}_i \rangle} y_i \langle \boldsymbol{w}^\star, \boldsymbol{x}_i \rangle}{\frac{1}{n} \sum_{i=1}^n e^{-y_i \langle \boldsymbol{w}, \boldsymbol{x}_i \rangle}} \geq \frac{\frac{1}{n} \sum_{i=1}^n e^{-y_i \langle \boldsymbol{w}, \boldsymbol{x}_i \rangle} \gamma^\star}{\frac{1}{n} \sum_{i=1}^n e^{-y_i \langle \boldsymbol{w}, \boldsymbol{x}_i \rangle}} = \gamma^\star,$$

$$\left\langle -\frac{\nabla \mathcal{L}(\boldsymbol{w})}{\mathcal{L}(\boldsymbol{w})}, \boldsymbol{w}^\star \right\rangle = \frac{\frac{1}{n} \sum_{i=1}^n e^{-y_i \langle \boldsymbol{w}, \boldsymbol{x}_i \rangle} y_i \langle \boldsymbol{w}^\star, \boldsymbol{x}_i \rangle}{\frac{1}{n} \sum_{i=1}^n e^{-y_i \langle \boldsymbol{w}, \boldsymbol{x}_i \rangle}} \leq \frac{\frac{1}{n} \sum_{i=1}^n e^{-y_i \langle \boldsymbol{w}, \boldsymbol{x}_i \rangle}}{\frac{1}{n} \sum_{i=1}^n e^{-y_i \langle \boldsymbol{w}, \boldsymbol{x}_i \rangle}} = 1.$$

For the lower bound of $\|\nabla\mathcal{L}(\boldsymbol{w})/\mathcal{L}(\boldsymbol{w})\|$, it holds that

$$\left\|\frac{\nabla\mathcal{L}(\boldsymbol{w})}{\mathcal{L}(\boldsymbol{w})}\right\| \geq \left\langle -\frac{\nabla\mathcal{L}(\boldsymbol{w})}{\mathcal{L}(\boldsymbol{w})}, \boldsymbol{w}^\star \right\rangle \geq \gamma^\star.$$

For the upper bound of $\|\nabla\mathcal{L}(\boldsymbol{w})/\mathcal{L}(\boldsymbol{w})\|$, it holds that

$$\left\|\frac{\nabla\mathcal{L}(\boldsymbol{w})}{\mathcal{L}(\boldsymbol{w})}\right\| = \left\|-\frac{\frac{1}{n}\sum_{i=1}^{n}e^{-y_i\langle\boldsymbol{w},\boldsymbol{x}_i\rangle}y_i\boldsymbol{x}_i}{\frac{1}{n}\sum_{i=1}^{n}e^{-y_i\langle\boldsymbol{w},\boldsymbol{x}_i\rangle}}\right\| \leq \frac{\frac{1}{n}\sum_{i=1}^{n}e^{-y_i\langle\boldsymbol{w},\boldsymbol{x}_i\rangle}\|y_i\boldsymbol{x}_i\|}{\frac{1}{n}\sum_{i=1}^{n}e^{-y_i\langle\boldsymbol{w},\boldsymbol{x}_i\rangle}} \leq 1.$$

$\square$

**Lemma C.2** ((Ji et al., 2020)). *Under Assumption 3.1, let $\boldsymbol{w}(t)$ be trained by GD (2) with $\eta \leq 1/2$ starting from $\boldsymbol{w}(0) = \boldsymbol{0}$, then GD converges to the max-margin direction: $\lim_{t\to+\infty}\frac{\boldsymbol{w}(t)}{\|\boldsymbol{w}(t)\|} \to \boldsymbol{w}^\star$.*

### C.1 PROOF OF THEOREM 6.2

*Proof of Theorem 6.2.*
Under Assumption 3.1, the $\ell_2$-max margin direction is unique, and we denote

$$\gamma^\star = \max_{\|\boldsymbol{w}\|\leq 1}\min_{i\in[n]} y_i\langle\boldsymbol{w},\boldsymbol{x}_i\rangle,$$
$$\boldsymbol{w}^\star = \arg\max_{\|\boldsymbol{w}\|\leq 1}\min_{i\in[n]} y_i\langle\boldsymbol{w},\boldsymbol{x}_i\rangle.$$

According Assumption 6.1, there exist constants $H, D, \mu > 0$ such that

$$\left\langle\frac{\nabla\mathcal{L}(\boldsymbol{w})}{\mathcal{L}(\boldsymbol{w})}, \frac{\mathcal{P}_\perp(\boldsymbol{w})}{\|\mathcal{P}_\perp(\boldsymbol{w})\|}\right\rangle \geq \mu \text{ holds for any } \boldsymbol{w} \in \mathbb{C}(D;H),$$

where

$$\mathbb{C}(D;H) := \left\{\boldsymbol{w}\in\mathbb{R}^d : \|\mathcal{P}_\perp(\boldsymbol{w})\| = D; \langle\boldsymbol{w},\boldsymbol{w}^\star\rangle \geq H\right\}$$

Analysis of Phase I.

Phase I is a warm-up phase. We will prove that at the end of this phase, trained $\boldsymbol{w}$ can be scaled onto $\mathbb{C}(D;H)$. First, we choose the error

$$\epsilon = \min\left\{\frac{D}{2H}, \frac{1}{2}\right\}.$$

With the help of Lemma C.2, we know that there exists $T^\epsilon$ such that $\left\|\frac{\boldsymbol{w}(T^\epsilon)}{\|\boldsymbol{w}(T^\epsilon)\|} - \boldsymbol{w}^\star\right\| < \epsilon$, which implies the inner satisfies:

$$\left\langle\frac{\boldsymbol{w}(T^\epsilon)}{\|\boldsymbol{w}(T^\epsilon)\|}, \boldsymbol{w}^\star\right\rangle = \frac{1}{2}\left(2 - \left\|\frac{\boldsymbol{w}(T^\epsilon)}{\|\boldsymbol{w}(T^\epsilon)\|} - \boldsymbol{w}^\star\right\|^2\right) > 1 - \frac{\epsilon^2}{2}.$$

Therefore, at $T^\epsilon$, it holds that:

$$\begin{aligned}
\frac{\|\mathcal{P}_\perp(\boldsymbol{w}(T^\epsilon))\|}{\langle\boldsymbol{w}(T^\epsilon),\boldsymbol{w}^\star\rangle} &= \frac{\|\boldsymbol{w}(T^\epsilon) - \mathcal{P}(\boldsymbol{w}(T^\epsilon))\|}{\langle\boldsymbol{w}(T^\epsilon),\boldsymbol{w}^\star\rangle} = \left\|\frac{\boldsymbol{w}(T^\epsilon)}{\langle\boldsymbol{w}(T^\epsilon),\boldsymbol{w}^\star\rangle} - \boldsymbol{w}^\star\right\| \\
&= \left\|\frac{\boldsymbol{w}(T^\epsilon)}{\langle\boldsymbol{w}(T^\epsilon)),\boldsymbol{w}^\star\rangle} - \boldsymbol{w}^\star\right\| = \left\|\frac{\boldsymbol{w}(T^\epsilon)}{\langle\boldsymbol{w}(T^\epsilon)),\boldsymbol{w}^\star\rangle} - \frac{\boldsymbol{w}(T^\epsilon)}{\|\boldsymbol{w}(T^\epsilon)\|} + \frac{\boldsymbol{w}(T^\epsilon)}{\|\boldsymbol{w}(T^\epsilon)\|} - \boldsymbol{w}^\star\right\| \\
&\leq \left\|\frac{\boldsymbol{w}(T^\epsilon)}{\langle\boldsymbol{w}(T^\epsilon)),\boldsymbol{w}^\star\rangle} - \frac{\boldsymbol{w}(T^\epsilon)}{\|\boldsymbol{w}(T^\epsilon)\|}\right\| + \left\|\frac{\boldsymbol{w}(T^\epsilon)}{\|\boldsymbol{w}(T^\epsilon)\|} - \boldsymbol{w}^\star\right\| < \left|\frac{\langle\boldsymbol{w}(T^\epsilon)),\boldsymbol{w}^\star\rangle - \|\boldsymbol{w}(T^\epsilon)\|}{\langle\boldsymbol{w}(T^\epsilon)),\boldsymbol{w}^\star\rangle}\right| + \epsilon \\
&= \left|1 - \frac{1}{\left\langle\frac{\boldsymbol{w}(T^\epsilon)}{\|\boldsymbol{w}(T^\epsilon)\|},\boldsymbol{w}^\star\right\rangle}\right| + \epsilon < \frac{1}{1-\frac{\epsilon^2}{2}} - 1 + \epsilon = \frac{\frac{\epsilon^2}{2}}{1-\frac{\epsilon^2}{2}} + \epsilon < \frac{4\epsilon^2}{7} + \epsilon
\end{aligned}$$

$$\leq \left(\frac{2}{7}+1\right)\epsilon \leq 2\epsilon \leq \min\left\{\frac{D}{H}, 1\right\}.$$

We choose $T^{\mathrm{I}} = T^{\epsilon} = \Theta(1)$, and we obtain $\boldsymbol{w}(T^{\mathrm{I}})$ at the end of Phase I.

Analysis of Phase II.

For simplicity, due to $T^{\mathrm{I}}$ is an constant, we replace the time $t$ to $t - T^{\mathrm{I}}$ in the proof of Phase II. This means that Phase II starts from $t = 0$ with the initialization $\boldsymbol{w}(0) \leftarrow \boldsymbol{w}(T^{\mathrm{I}})$.

In this proof, we choose

$$\eta = \mu D,\ T_k = 2k,\ R_k = \frac{D\|\boldsymbol{w}(T_k)\|}{\|\mathcal{P}_{\perp}(\boldsymbol{w}(T_k))\|},\ \forall k \geq 0.$$

Recalling Algorithm 1, the update rule is:

$$\cdots ;$$
$$\boldsymbol{w}(2k+1) = R_k \frac{\boldsymbol{w}(2k)}{\|\boldsymbol{w}(2k)\|};$$
$$\boldsymbol{v}(2k+2) = \boldsymbol{w}(2k+1) - \eta \frac{\nabla\mathcal{L}(\boldsymbol{w}(2k+1))}{\mathcal{L}(\boldsymbol{w}(2k+1))};$$
$$\boldsymbol{w}(2k+2) = \mathrm{Proj}_{\mathbb{B}(0,\|\boldsymbol{w}(2k+1)\|)}\left(\boldsymbol{v}(2k+2)\right);$$
$$\boldsymbol{w}(2k+3) = R_{k+1}\frac{\boldsymbol{w}(2k+2)}{\|\boldsymbol{w}(2k+2)\|};$$
$$\cdots$$

In general, we aim to prove the following statements:

(S1). $\boldsymbol{w}(2k+1) \in \mathbb{C}(D; H),\ \forall k \geq 0.$

(S2). $\langle \boldsymbol{w}(2k+1), \boldsymbol{w}^{\star}\rangle \geq \dfrac{1}{\left(\sqrt{1-2\mu}\right)^k}\left(\langle \boldsymbol{w}(1), \boldsymbol{w}^{\star}\rangle + \dfrac{\gamma^{\star}}{1-\sqrt{1-2\mu}}\right) - \dfrac{\gamma^{\star}}{1-\sqrt{1-2\mu}},\ \forall k \geq 0;$

$$\langle \boldsymbol{w}(2k+1), \boldsymbol{w}^{\star}\rangle \leq \dfrac{1}{\left(\sqrt{1-\mu^2}\right)^k}\left(\langle \boldsymbol{w}(1), \boldsymbol{w}^{\star}\rangle + \dfrac{1}{1-\sqrt{1-\mu^2}}\right) - \dfrac{1}{1-\sqrt{1-\mu^2}},\ \forall k \geq 0.$$

(S3). $D\sqrt{1-2\mu} \leq \|\mathcal{P}_{\perp}(\boldsymbol{v}(2k+2))\| \leq D\sqrt{1-\mu^2},\ \forall k \geq 0.$

(S4). $\dfrac{\langle \boldsymbol{w}(2k+2), \boldsymbol{w}^{\star}\rangle}{\|\mathcal{P}_{\perp}(\boldsymbol{w}(2k+2))\|} = \dfrac{\langle \boldsymbol{w}(2k+3), \boldsymbol{w}^{\star}\rangle}{\|\mathcal{P}_{\perp}(\boldsymbol{w}(2k+3))\|} = \dfrac{\langle \boldsymbol{w}(1), \boldsymbol{w}^{\star}\rangle}{D}e^{\Theta(k)}.$

(S5). $R_{k+1} = \langle \boldsymbol{w}(1), \boldsymbol{w}^{\star}\rangle e^{\Theta(k)}.$

(S6). $\left\|\dfrac{\boldsymbol{w}(t)}{\|\boldsymbol{w}(t)\|} - \boldsymbol{w}^{\star}\right\| = \dfrac{D}{\langle \boldsymbol{w}(1), \boldsymbol{w}^{\star}\rangle}e^{-\Theta(t)}.$

(S7). $\gamma^{\star} - \gamma(\boldsymbol{w}(t)) = \dfrac{D}{\langle \boldsymbol{w}(1), \boldsymbol{w}^{\star}\rangle}e^{-\Theta(t)}.$

Step I. Proof of (S1)(S2).

In this step, we will prove (S1)(S2) by induction.

Step I (i). We prove (S1)(S2) for $k = 0$. Recalling our analysis of Phase I, it holds that

$$\frac{\|\mathcal{P}_{\perp}(\boldsymbol{w}(0))\|}{\langle \boldsymbol{w}(0)), \boldsymbol{w}^{\star}\rangle} \leq \min\left\{\frac{D}{H}, 1\right\}.$$

Thus, if we choose $R_0 = \frac{D\|\boldsymbol{w}(0)\|}{\|\mathcal{P}_{\perp}(\boldsymbol{w}(0))\|}$ in Algorithm 1, then $\boldsymbol{w}(1) = \frac{D}{\|\mathcal{P}_{\perp}(\boldsymbol{w}(0))\|} \cdot \boldsymbol{w}(0)$ and $\boldsymbol{w}(1)$ satisfies:

$$\|\mathcal{P}_{\perp}(\boldsymbol{w}(1))\| = \left\|\mathcal{P}_{\perp}\left(\frac{D}{\|\mathcal{P}_{\perp}(\boldsymbol{w}(0))\|}\boldsymbol{w}(0)\right)\right\| = \left\|\frac{D\mathcal{P}_{\perp}(\boldsymbol{w}(0))}{\|\mathcal{P}_{\perp}(\boldsymbol{w}(0))\|}\right\| = D;$$

$$\langle \boldsymbol{w}(1), \boldsymbol{w}^\star \rangle = \left\langle \frac{D}{\|\mathcal{P}_\perp(\boldsymbol{w}(0))\|} \boldsymbol{w}(0), \boldsymbol{w}^\star \right\rangle = D \left\langle \frac{\boldsymbol{w}(0)}{\|\mathcal{P}_\perp(\boldsymbol{w}(0))\|}, \boldsymbol{w}^\star \right\rangle$$

$$= D \frac{\langle \boldsymbol{w}(0), \boldsymbol{w}^\star \rangle}{\|\mathcal{P}_\perp(\boldsymbol{w}(0))\|} \geq \frac{D}{\min\left\{\frac{D}{H}, 1\right\}} = \max\{H, D\}.$$

which means that (S1) $\boldsymbol{w}(1) \in \mathbb{C}(D; H)$ holds for $k = 0$. As for (S2), it is trivial for $k = 0$.

Step I (ii). Assume (S1)(S2) hold for any $0 \leq k' \leq k$. Then we will prove for $k' = k + 1$.

First, it is easy to bound the difference between $\langle \boldsymbol{v}(2k + 2), \boldsymbol{w}^\star \rangle$ and $\langle \boldsymbol{w}(2k + 1), \boldsymbol{w}^\star \rangle$:

$$\langle \boldsymbol{v}(2k + 2), \boldsymbol{w}^\star \rangle - \langle \boldsymbol{w}(2k + 1), \boldsymbol{w}^\star \rangle$$
$$= \eta \left\langle -\frac{\nabla \mathcal{L}(\boldsymbol{w}(2k + 1))}{\mathcal{L}(\boldsymbol{w}(2k + 1))}, \boldsymbol{w}^\star \right\rangle \overset{\text{Lemma C.1}}{\in} [\eta \gamma^\star, \eta] = [\mu \gamma^\star D, \mu D]. \tag{5}$$

Secondly, notice the following fact about $\boldsymbol{w}(2k + 3)$:

$$\boldsymbol{w}(2k + 3) = R_{k+1} \frac{\boldsymbol{w}(2k + 2)}{\|\boldsymbol{w}(2k + 2)\|} = \frac{D \|\boldsymbol{w}(2k + 2)\|}{\|\mathcal{P}_\perp(\boldsymbol{w}(2k + 2))\|} \frac{\boldsymbol{w}(2k + 2)}{\|\boldsymbol{w}(2k + 2)\|}$$
$$= \frac{D}{\|\mathcal{P}_\perp(\boldsymbol{w}(2k + 2))\|} \boldsymbol{w}(2k + 2) = \frac{D}{\|\mathcal{P}_\perp(\boldsymbol{v}(2k + 2))\|} \boldsymbol{v}(2k + 2). \tag{6}$$

With the help of the estimates above and the induction, now we can give the following two-sided bounds for $\langle \boldsymbol{w}(2k + 3), \boldsymbol{w}^\star \rangle$.

- Upper bound for $\langle \boldsymbol{w}(2k + 3), \boldsymbol{w}^\star \rangle$:

$$\langle \boldsymbol{w}(2k + 3), \boldsymbol{w}^\star \rangle \overset{(6)}{=} D \frac{\langle \boldsymbol{v}(2k + 2), \boldsymbol{w}^\star \rangle}{\|\mathcal{P}_\perp(\boldsymbol{v}(2k + 2))\|}$$
$$\overset{(5)}{\leq} D \frac{\langle \boldsymbol{w}(2k + 1), \boldsymbol{w}^\star \rangle + 1}{\sqrt{1 - \mu^2} D} = \frac{\langle \boldsymbol{w}(2k + 1), \boldsymbol{w}^\star \rangle + 1}{\sqrt{1 - \mu^2}}$$
$$\overset{\text{induction}}{\leq} \frac{1}{\sqrt{1 - \mu^2}} \left( \frac{1}{\left(\sqrt{1 - \mu^2}\right)^k} \left( \langle \boldsymbol{w}(1), \boldsymbol{w}^\star \rangle + \frac{1}{1 - \sqrt{1 - \mu^2}} \right) - \frac{1}{1 - \sqrt{1 - \mu^2}} + 1 \right) \tag{7}$$
$$= \frac{1}{\left(\sqrt{1 - \mu^2}\right)^{k+1}} \left( \langle \boldsymbol{w}(1), \boldsymbol{w}^\star \rangle + \frac{1}{1 - \sqrt{1 - \mu^2}} \right) - \frac{1}{1 - \sqrt{1 - \mu^2}}.$$

- Lower bound for $\langle \boldsymbol{w}(2k + 3), \boldsymbol{w}^\star \rangle$:

$$\langle \boldsymbol{w}(2k + 3), \boldsymbol{w}^\star \rangle \overset{(6)}{=} D \frac{\langle \boldsymbol{v}(2k + 2), \boldsymbol{w}^\star \rangle}{\|\mathcal{P}_\perp(\boldsymbol{v}(2k + 2))\|}$$
$$\overset{(5)}{\geq} D \frac{\langle \boldsymbol{w}(2k + 1), \boldsymbol{w}^\star \rangle + \gamma^\star}{\sqrt{1 - 2\mu} D} = \frac{\langle \boldsymbol{w}(2k + 1), \boldsymbol{w}^\star \rangle + \gamma^\star}{\sqrt{1 - 2\mu}}$$
$$\overset{\text{induction}}{\geq} \frac{1}{\sqrt{1 - 2\mu}} \left( \frac{1}{\left(\sqrt{1 - 2\mu}\right)^k} \left( \langle \boldsymbol{w}(1), \boldsymbol{w}^\star \rangle + \frac{\gamma^\star}{1 - \sqrt{1 - 2\mu}} \right) - \frac{\gamma^\star}{1 - \sqrt{1 - 2\mu}} + \gamma^\star \right) \tag{8}$$
$$= \frac{1}{\left(\sqrt{1 - 2\mu}\right)^{k+1}} \left( \langle \boldsymbol{w}(1), \boldsymbol{w}^\star \rangle + \frac{\gamma^\star}{1 - \sqrt{1 - 2\mu}} \right) - \frac{\gamma^\star}{1 - \sqrt{1 - 2\mu}}.$$

Hence, from (7)(8), we have proved that (S2) holds for $k + 1$.

Moreover, we have the following facts:

$$\|\mathcal{P}_\perp(\boldsymbol{w}(2k + 3))\| \overset{(6)}{=} \left\| \mathcal{P}_\perp \left( \frac{D}{\|\mathcal{P}_\perp(\boldsymbol{v}(2k + 2))\|} \boldsymbol{v}(2k + 2) \right) \right\|$$

$$= \left\| D \frac{\mathcal{P}_\perp(\boldsymbol{v}(2k+2))}{\|\mathcal{P}_\perp(\boldsymbol{v}(2k+2))\|} \right\| = D,$$

$$\langle \boldsymbol{w}(2k+3), \boldsymbol{w}^\star \rangle \overset{(8)}{\geq} \frac{1}{\left(\sqrt{1-2\mu}\right)^{k+1}} \left( \langle \boldsymbol{w}(1), \boldsymbol{w}^\star \rangle + \frac{\gamma^\star}{1-\sqrt{1-2\mu}} \right) - \frac{\gamma^\star}{1-\sqrt{1-2\mu}}$$

$$\geq \frac{\langle \boldsymbol{w}(1), \boldsymbol{w}^\star \rangle}{\left(\sqrt{1-2\mu}\right)^{k+1}} \geq \langle \boldsymbol{w}(1), \boldsymbol{w}^\star \rangle \geq H;$$

which means that (S1) holds for $k+1$, i.e., $\boldsymbol{w}(2k+3) \in \mathbb{C}(D; H)$.

Now we have proved (S1)(S2) for any $k \geq 0$ by induction.

Step II. Proof of (S3).

In this step, we will prove (S3) directly. For any $k \geq 0$, we can derive the following two-sides bounds:

- For the upper bound of $\|\mathcal{P}_\perp(\boldsymbol{v}(2k+2))\|$, we have:

$$\|\mathcal{P}_\perp(\boldsymbol{v}(2k+2))\|^2 = \left\| \mathcal{P}_\perp(\boldsymbol{w}(2k+1)) - \eta \mathcal{P}_\perp\left( \frac{\nabla\mathcal{L}(\boldsymbol{w}(2k+1))}{\mathcal{L}(\boldsymbol{w}(2k+1))} \right) \right\|^2$$

$$= \|\mathcal{P}_\perp(\boldsymbol{w}(2k+1))\|^2 + \eta^2 \left\| \mathcal{P}_\perp\left( \frac{\nabla\mathcal{L}(\boldsymbol{w}(2k+1))}{\mathcal{L}(\boldsymbol{w}(2k+1))} \right) \right\|^2$$

$$- 2\eta \left\langle \mathcal{P}_\perp(\boldsymbol{w}(2k+1)), \mathcal{P}_\perp\left( \frac{\nabla\mathcal{L}(\boldsymbol{w}(2k+1))}{\mathcal{L}(\boldsymbol{w}(2k+1))} \right) \right\rangle$$

$$= D^2 + \eta^2 \left\| \mathcal{P}_\perp\left( \frac{\nabla\mathcal{L}(\boldsymbol{w}(2k+1))}{\mathcal{L}(\boldsymbol{w}(2k+1))} \right) \right\|^2 - 2\eta D \left\langle \frac{\mathcal{P}_\perp(\boldsymbol{w}(2k+1))}{\|\mathcal{P}_\perp(\boldsymbol{w}(2k+1))\|}, \frac{\nabla\mathcal{L}(\boldsymbol{w}(2k+1))}{\mathcal{L}(\boldsymbol{w}(2k+1))} \right\rangle$$

$$\leq D^2 + \eta^2 \left\| \frac{\nabla\mathcal{L}(\boldsymbol{w}(2k+1))}{\mathcal{L}(\boldsymbol{w}(2k+1))} \right\|^2 - 2\eta D \left\langle \frac{\mathcal{P}_\perp(\boldsymbol{w}(2k+1))}{\|\mathcal{P}_\perp(\boldsymbol{w}(2k+1))\|}, \frac{\nabla\mathcal{L}(\boldsymbol{w}(2k+1))}{\mathcal{L}(\boldsymbol{w}(2k+1))} \right\rangle$$

$$\overset{\text{Lemma C.1}}{\leq} D^2 + \eta^2 - 2\eta D\mu = D^2 + \mu^2 D^2 - 2\mu^2 D^2 = (1-\mu^2)D^2.$$

$$(9)$$

- For the lower bound of $\|\mathcal{P}_\perp(\boldsymbol{v}(2k+2))\|$, we have:

$$\|\mathcal{P}_\perp(\boldsymbol{v}(2k+2))\|^2 = \left\| \mathcal{P}_\perp(\boldsymbol{w}(2k+1)) - \eta \mathcal{P}_\perp\left( \frac{\nabla\mathcal{L}(\boldsymbol{w}(2k+1))}{\mathcal{L}(\boldsymbol{w}(2k+1))} \right) \right\|^2$$

$$= D^2 + \eta^2 \left\| \mathcal{P}_\perp\left( \frac{\nabla\mathcal{L}(\boldsymbol{w}(2k+1))}{\mathcal{L}(\boldsymbol{w}(2k+1))} \right) \right\|^2$$

$$- 2\eta D \left\langle \frac{\mathcal{P}_\perp(\boldsymbol{w}(2k+1))}{\|\mathcal{P}_\perp(\boldsymbol{w}(2k+1))\|}, \frac{\nabla\mathcal{L}(\boldsymbol{w}(2k+1))}{\mathcal{L}(\boldsymbol{w}(2k+1))} \right\rangle$$

$$\geq D^2 - 2\eta D \left\langle \frac{\mathcal{P}_\perp(\boldsymbol{w}(2k+1))}{\|\mathcal{P}_\perp(\boldsymbol{w}(2k+1))\|}, \frac{\nabla\mathcal{L}(\boldsymbol{w}(2k+1))}{\mathcal{L}(\boldsymbol{w}(2k+1))} \right\rangle$$

$$\geq D^2 - 2\eta D \left\| \frac{\nabla\mathcal{L}(\boldsymbol{w}(2k+1))}{\mathcal{L}(\boldsymbol{w}(2k+1))} \right\|$$

$$\overset{\text{Lemma C.1}}{\geq} D^2 - 2\eta D = D^2 - 2\mu D^2 = (1-2\mu)D^2.$$

$$(10)$$

Hence, we have proved (S3).

Step III. Proof of (S4)(S5)(S6).

First, we derive two-sided bounds for $\frac{\langle \boldsymbol{w}(2k+2), \boldsymbol{w}^\star \rangle}{\|\mathcal{P}_\perp(\boldsymbol{w}(2k+2))\|}$. For any $k \geq 0$, we have:

- Upper bound of $\frac{\langle \boldsymbol{w}(2k+2), \boldsymbol{w}^\star \rangle}{\|\mathcal{P}_\perp(\boldsymbol{w}(2k+2))\|}$.

$$
\begin{aligned}
\frac{\langle \boldsymbol{w}(2k+2), \boldsymbol{w}^\star \rangle}{\|\mathcal{P}_\perp(\boldsymbol{w}(2k+2))\|} &= \frac{\langle \boldsymbol{v}(2k+2), \boldsymbol{w}^\star \rangle}{\|\mathcal{P}_\perp(\boldsymbol{v}(2k+2))\|} \\
&\overset{(5)}{\leq} \frac{\langle \boldsymbol{w}(2k+1), \boldsymbol{w}^\star \rangle + \mu D}{\|\mathcal{P}_\perp(\boldsymbol{v}(2k+2))\|} \\
&\overset{(S2)(S3)}{\leq} \frac{\frac{1}{\left(\sqrt{1-\mu^2}\right)^k}\left(\langle \boldsymbol{w}(1), \boldsymbol{w}^\star \rangle + \frac{1}{1-\sqrt{1-\mu^2}}\right) - \frac{1}{1-\sqrt{1-\mu^2}} + \mu D}{D\sqrt{1-2\mu}} \\
&\leq \frac{\langle \boldsymbol{w}(1), \boldsymbol{w}^\star \rangle}{D} e^{\Theta(k)}.
\end{aligned}
$$

- Lower bound of $\frac{\langle \boldsymbol{w}(2k+2), \boldsymbol{w}^\star \rangle}{\|\mathcal{P}_\perp(\boldsymbol{w}(2k+2))\|}$.

$$
\begin{aligned}
\frac{\langle \boldsymbol{w}(2k+2), \boldsymbol{w}^\star \rangle}{\|\mathcal{P}_\perp(\boldsymbol{w}(2k+2))\|} &= \frac{\langle \boldsymbol{v}(2k+2), \boldsymbol{w}^\star \rangle}{\|\mathcal{P}_\perp(\boldsymbol{v}(2k+2))\|} \\
&\overset{(5)}{\geq} \frac{\langle \boldsymbol{w}(2k+1), \boldsymbol{w}^\star \rangle + \mu\gamma^\star D}{\|\mathcal{P}_\perp(\boldsymbol{v}(2k+2))\|} \\
&\overset{(S2)(S3)}{\geq} \frac{\frac{1}{\left(\sqrt{1-2\mu}\right)^k}\left(\langle \boldsymbol{w}(1), \boldsymbol{w}^\star \rangle + \frac{\gamma^\star}{1-\sqrt{1-2\mu}}\right) - \frac{\gamma^\star}{1-\sqrt{1-2\mu}} + \mu\gamma^\star D}{(1-\mu)D} \\
&\geq \frac{\langle \boldsymbol{w}(1), \boldsymbol{w}^\star \rangle}{D} e^{\Theta(k)}.
\end{aligned}
$$

Additionally, notice

$$
\frac{\langle \boldsymbol{w}(2k+2), \boldsymbol{w}^\star \rangle}{\|\mathcal{P}_\perp(\boldsymbol{w}(2k+2))\|} = \frac{\langle \boldsymbol{w}(2k+3), \boldsymbol{w}^\star \rangle}{\|\mathcal{P}_\perp(\boldsymbol{w}(2k+3))\|},
$$

we obtain (S4):

$$
\frac{\langle \boldsymbol{w}(2k+2), \boldsymbol{w}^\star \rangle}{\|\mathcal{P}_\perp(\boldsymbol{w}(2k+2))\|} = \frac{\langle \boldsymbol{w}(2k+3), \boldsymbol{w}^\star \rangle}{\|\mathcal{P}_\perp(\boldsymbol{w}(2k+3))\|} = \frac{\langle \boldsymbol{w}(1), \boldsymbol{w}^\star \rangle}{D} e^{\Theta(k)}.
$$

Furthermore, Combining (S4) and the following fact

$$
\begin{aligned}
R_{k+1} &= \frac{D\|\boldsymbol{w}(2k+2)\|}{\|\mathcal{P}_\perp(\boldsymbol{w}(2k+2))\|} = \frac{D\|\boldsymbol{v}(2k+2)\|}{\|\mathcal{P}_\perp(\boldsymbol{v}(2k+2))\|} \\
&= D\frac{\sqrt{\langle \boldsymbol{v}(2k+2), \boldsymbol{w}^\star \rangle^2 + \|\mathcal{P}_\perp(\boldsymbol{v}(2k+2))\|^2}}{\|\mathcal{P}_\perp(\boldsymbol{v}(2k+2))\|} = D\sqrt{\frac{\langle \boldsymbol{v}(2k+2), \boldsymbol{w}^\star \rangle^2}{\|\mathcal{P}_\perp(\boldsymbol{v}(2k+2))\|^2} + 1},
\end{aligned}
$$

we can obtain (S5):

$$
R_{k+1} = \langle \boldsymbol{w}(1), \boldsymbol{w}^\star \rangle e^{\Theta(k)}.
$$

In the same way, we can prove

$$
\begin{aligned}
\left\| \frac{\boldsymbol{w}(2k)}{\|\boldsymbol{w}(2k)\|} - \boldsymbol{w}^\star \right\| &= \left\| \frac{\boldsymbol{w}(2k+1)}{\|\boldsymbol{w}(2k+1)\|} - \boldsymbol{w}^\star \right\| = 2\left(1 - \left\langle \frac{\boldsymbol{w}(2k+1)}{\|\boldsymbol{w}(2k+1)\|}, \boldsymbol{w}^\star \right\rangle\right) \\
&= 2\left(1 - \frac{\langle \boldsymbol{w}(2k+1), \boldsymbol{w}^\star \rangle}{\|\boldsymbol{w}(2k+1)\|}\right) = 2\left(1 - \frac{\langle \boldsymbol{w}(2k+1), \boldsymbol{w}^\star \rangle}{\sqrt{\langle \boldsymbol{w}(2k+1), \boldsymbol{w}^\star \rangle^2 + \|\mathcal{P}_\perp(\boldsymbol{w}(2k+1))\|^2}}\right) \\
&= 2\left(1 - \frac{1}{\sqrt{1 + \frac{\|\mathcal{P}_\perp(\boldsymbol{w}(2k+1))\|^2}{\langle \boldsymbol{w}(2k+1), \boldsymbol{w}^\star \rangle^2}}}\right) \overset{(S4)}{=} 2\left(1 - \frac{1}{\sqrt{1 + \frac{D^2}{\langle \boldsymbol{w}(1), \boldsymbol{w}^\star \rangle^2} e^{-\Theta(k)}}}\right)
\end{aligned}
$$

$$= \frac{D}{\langle \boldsymbol{w}(1), \boldsymbol{w}^\star \rangle} e^{-\Theta(k)},$$

which means (S6):

$$\left\| \frac{\boldsymbol{w}(t)}{\|\boldsymbol{w}(t)\|} - \boldsymbol{w}^\star \right\| = \frac{D}{\langle \boldsymbol{w}(1), \boldsymbol{w}^\star \rangle} e^{-\Theta(t)},$$

Step III. Proof of (S7). Using Lemma A.4 and (S6), we obtain (S7).

Conclusions.

From our proof of Phase II, we have $\langle \boldsymbol{w}(1), \boldsymbol{w}^\star \rangle \geq \max\{H, D\}$. Taking this fact into (S6)(S7), we obtain our conclusions:

$$\left\| \frac{\boldsymbol{w}(t)}{\|\boldsymbol{w}(t)\|} - \boldsymbol{w}^\star \right\| = e^{-\Omega(t)};$$
$$\gamma^\star - \gamma(\boldsymbol{w}(t)) = e^{-\Omega(t)}.$$

$\square$

*Proof of Theorem 6.3.*
Notice that Assumption 5.4 can imply Assumption 6.1. Therefore, for the case of GD (Phase I) + PRGD (Phase II), this theorem is the direct corollary of Theorem 6.2.

Then we discuss the other case: NGD (Phase I) + PRGD (Phase II).

$\square$

## C.2 PROOF OF THEOREM 6.4

*Proof for Theorem 6.4.*
As the representative, we first provide the detailed proof for NGD, which is more difficult to analyze.

Without loss of generality, we can assume $\text{span}\{\boldsymbol{x}_1, \cdots, \boldsymbol{x}_n\} = \mathbb{R}^d$. This is because: GD, NGD, and PRGD can only evaluate in $\text{span}\{\boldsymbol{x}_i : i \in [n]\}$, i.e. $\boldsymbol{w}(t) \in \text{span}\{\boldsymbol{x}_i : i \in [n]\}$. If $\text{span}\{\boldsymbol{x}_1, \cdots, \boldsymbol{x}_n\} \neq \mathbb{R}^d$, we only need to change the proof in the subspace $\text{span}\{\boldsymbol{x}_1, \cdots, \boldsymbol{x}_n\}$.

With the help of Theorem C.3, the upper bounds hold: $\left\| \frac{\boldsymbol{w}(t)}{\|\boldsymbol{w}(t)\|} - \boldsymbol{w}^\star \right\| = \mathcal{O}(1/t)$. So we only need to prove the lower bounds:

$$\left\| \frac{\boldsymbol{w}(t)}{\|\boldsymbol{w}(t)\|} - \boldsymbol{w}^\star \right\| = \Omega(1/t).$$

For simplicity, we denote the optimization problem orthogonal to $\boldsymbol{w}^\star$ as

$$\min_{\boldsymbol{v}} : \mathcal{L}_\perp(\boldsymbol{v}) = \frac{1}{|\mathcal{I}|} \sum_{i \in \mathcal{I}} \exp\left(-y_i \langle \boldsymbol{v}, \mathcal{P}_\perp(\boldsymbol{x}_i) \rangle\right), \boldsymbol{v} \in \text{span}\{\mathcal{P}_\perp(\boldsymbol{x}_i) : i \in \mathcal{I}\}.$$

In this proof, we focus on the dynamics of $\mathcal{P}_\perp(\boldsymbol{w}(t))$, satisfying:

$$\mathcal{P}_\perp(\boldsymbol{w}(t+1)) = \mathcal{P}_\perp(\boldsymbol{w}(t)) - \eta \mathcal{P}_\perp\left(\frac{\nabla \mathcal{L}(\boldsymbol{w})}{\mathcal{L}(\boldsymbol{w})}\right)$$

$$= \mathcal{P}_\perp(\boldsymbol{w}(t)) - \eta \mathcal{P}_\perp\left(\frac{\frac{1}{n}\sum_{i=1}^n e^{-\langle \boldsymbol{w}(t), \boldsymbol{x}_i y_i \rangle} \boldsymbol{x}_i y_i}{\frac{1}{n}\sum_{i=1}^n e^{-\langle \boldsymbol{w}(t), \boldsymbol{x}_i y_i \rangle}}\right)$$

$$= \mathcal{P}_\perp(\boldsymbol{w}(t)) - \eta \frac{\sum_{i=1}^n e^{-\langle \boldsymbol{w}(t), \boldsymbol{x}_i y_i \rangle} \mathcal{P}_\perp(\boldsymbol{x}_i y_i)}{\sum_{i=1}^n e^{-\langle \boldsymbol{w}(t), \boldsymbol{x}_i y_i \rangle}}$$

Step I. The error of gradient in each step.

On the one hand, Theorem C.4 (iii) ensures that there exists an absolute constant $C > 0$ such that $\|\mathcal{P}_\perp(\boldsymbol{w}(t)) - \boldsymbol{v}^\star\| \leq C, \ \forall t$. On the other hand, Theorem C.3 shows that $\left\|\frac{\boldsymbol{w}(t)}{\|\boldsymbol{w}(t)\|} - \boldsymbol{w}^\star\right\| = \mathcal{O}(1/t)$ and $\|\boldsymbol{w}(t)\| = \Theta(t)$. And we notice the decomposition $\boldsymbol{w}(t) = \langle \boldsymbol{w}(t), \boldsymbol{w}^\star\rangle \boldsymbol{w}^\star + \mathcal{P}_\perp(\boldsymbol{w}(t))$.

Therefore, for any $\epsilon > 0$, there exists $T_\epsilon > 0$ such that for any $t > T_\epsilon$,

$(i).\ (1-\epsilon)\sum_{i\in\mathcal{I}} e^{-\langle \boldsymbol{w}(t),\boldsymbol{w}^\star\rangle\gamma^\star} \leq \sum_{i=1}^n e^{-\langle \boldsymbol{w}(t),\boldsymbol{x}_i y_i\rangle} \leq (1+\epsilon)\sum_{i\in\mathcal{I}} e^{-\langle \boldsymbol{w}(t),\boldsymbol{w}^\star\rangle\gamma^\star};$

$(ii).\ \left\|\sum_{i=1}^n e^{-\langle \boldsymbol{w}(t),\boldsymbol{x}_i y_i\rangle}\mathcal{P}_\perp(\boldsymbol{x}_i y_i) - \sum_{i\in\mathcal{I}} e^{-\langle \boldsymbol{w}(t),\boldsymbol{x}_i y_i\rangle}\mathcal{P}_\perp(\boldsymbol{x}_i y_i)\right\| \leq \epsilon\left\|\sum_{i=1}^n e^{-\langle \boldsymbol{w}(t),\boldsymbol{x}_i y_i\rangle}\mathcal{P}_\perp(\boldsymbol{x}_i y_i)\right\|.$

Consequently, (i)(ii) ensures that

$$
\left\|\mathcal{P}_\perp\left(\frac{\nabla\mathcal{L}(\boldsymbol{w})}{\mathcal{L}(\boldsymbol{w})}\right) - \nabla\mathcal{L}_\perp(\mathcal{P}_\perp(\boldsymbol{w}(t)))\right\|
$$

$$
= \left\|\frac{\sum_{i=1}^n e^{-\langle \boldsymbol{w}(t),\boldsymbol{x}_i y_i\rangle}\mathcal{P}_\perp(\boldsymbol{x}_i y_i)}{\sum_{i=1}^n e^{-\langle \boldsymbol{w}(t),\boldsymbol{x}_i y_i\rangle}} - \frac{1}{|\mathcal{I}|}\sum_{i\in\mathcal{I}} e^{-\langle \mathcal{P}_\perp(\boldsymbol{w}(t)),\mathcal{P}_\perp(\boldsymbol{x}_i y_i)\rangle}\mathcal{P}_\perp(\boldsymbol{x}_i y_i)\right\|
$$

$$
= \left\|\frac{\sum_{i=1}^n e^{-\langle \boldsymbol{w}(t),\boldsymbol{x}_i y_i\rangle}\mathcal{P}_\perp(\boldsymbol{x}_i y_i)}{\sum_{i=1}^n e^{-\langle \boldsymbol{w}(t),\boldsymbol{x}_i y_i\rangle}} - \frac{1}{|\mathcal{I}|}\sum_{i\in\mathcal{I}} e^{-\langle \mathcal{P}_\perp(\boldsymbol{w}(t)),\boldsymbol{x}_i y_i\rangle}\mathcal{P}_\perp(\boldsymbol{x}_i y_i)\right\|
$$

$$
= \left\|\frac{\sum_{i=1}^n e^{-\langle \boldsymbol{w}(t),\boldsymbol{x}_i y_i\rangle}\mathcal{P}_\perp(\boldsymbol{x}_i y_i)}{\sum_{i=1}^n e^{-\langle \boldsymbol{w}(t),\boldsymbol{x}_i y_i\rangle}} - \frac{\sum_{i\in\mathcal{I}} e^{-\langle \boldsymbol{w}(t),\boldsymbol{x}_i y_i\rangle}\mathcal{P}_\perp(\boldsymbol{x}_i y_i)}{\sum_{i\in\mathcal{I}} e^{-\langle \boldsymbol{w}(t),\boldsymbol{w}^\star\rangle\gamma^\star}}\right\|
$$

$$
\leq \left\|\frac{\sum_{i=1}^n e^{-\langle \boldsymbol{w}(t),\boldsymbol{x}_i y_i\rangle}\mathcal{P}_\perp(\boldsymbol{x}_i y_i)}{\sum_{i=1}^n e^{-\langle \boldsymbol{w}(t),\boldsymbol{x}_i y_i\rangle}} - \frac{\sum_{i\in\mathcal{I}} e^{-\langle \boldsymbol{w}(t),\boldsymbol{x}_i y_i\rangle}\mathcal{P}_\perp(\boldsymbol{x}_i y_i)}{\sum_{i=1}^n e^{-\langle \boldsymbol{w}(t),\boldsymbol{x}_i y_i\rangle}}\right\|
$$

$$
+ \left\|\frac{\sum_{i\in\mathcal{I}} e^{-\langle \boldsymbol{w}(t),\boldsymbol{x}_i y_i\rangle}\mathcal{P}_\perp(\boldsymbol{x}_i y_i)}{\sum_{i=1}^n e^{-\langle \boldsymbol{w}(t),\boldsymbol{x}_i y_i\rangle}} - \frac{\sum_{i\in\mathcal{I}} e^{-\langle \boldsymbol{w}(t),\boldsymbol{x}_i y_i\rangle}\mathcal{P}_\perp(\boldsymbol{x}_i y_i)}{\sum_{i\in\mathcal{I}} e^{-\langle \boldsymbol{w}(t),\boldsymbol{w}^\star\rangle\gamma^\star}}\right\|
$$

$$
\leq \epsilon\frac{\left\|\sum_{i=1}^n e^{-\langle \boldsymbol{w}(t),\boldsymbol{x}_i y_i\rangle}\mathcal{P}_\perp(\boldsymbol{x}_i y_i)\right\|}{\sum_{i=1}^n e^{-\langle \boldsymbol{w}(t),\boldsymbol{x}_i y_i\rangle}} + \epsilon\frac{\left\|\sum_{i\in\mathcal{I}} e^{-\langle \boldsymbol{w}(t),\boldsymbol{x}_i y_i\rangle}\mathcal{P}_\perp(\boldsymbol{x}_i y_i)\right\|}{\sum_{i=1}^n e^{-\langle \boldsymbol{w}(t),\boldsymbol{x}_i y_i\rangle}} \leq \epsilon + \epsilon = 2\epsilon.
$$

Step II. $\mathcal{P}_\perp(\boldsymbol{w}(t))$ can enter $\mathbb{B}(\boldsymbol{v}^\star; \delta)$ with any $\delta > 0$.

For any fixed $\delta > 0$, we assume that $\mathcal{P}_\perp(\boldsymbol{w}(t))$ can not enter $\mathbb{B}(\boldsymbol{v}^\star; \delta)$, which means $\mathcal{P}_\perp(\boldsymbol{w}(t)) \in \mathbb{B}(\boldsymbol{v}^\star; C) - \mathbb{B}(\boldsymbol{v}^\star; \delta)$ forever.

Recalling Theorem C.4, it ensures that $\mathcal{L}_\perp(\cdot)$ is $\mu$-strongly convex in $\mathbb{B}(\boldsymbol{v}^\star; C) - \mathbb{B}(\boldsymbol{v}^\star; \delta)$ for some $\mu > 0$. Therefore,

$$
\|\nabla\mathcal{L}_\perp(\boldsymbol{w})\| \geq \mu\|\boldsymbol{v} - \boldsymbol{v}^\star\| \geq \mu\delta, \quad \forall\boldsymbol{v} \in \mathbb{B}(\boldsymbol{v}^\star; C) - \mathbb{B}(\boldsymbol{v}^\star; \delta).
$$

If we select $\epsilon = \frac{\mu\delta}{100}$, the result in Step I ensures that there exists time $T_\epsilon > 0$ such that for any $t > T_\epsilon$,

$$
\left\|\mathcal{P}_\perp\left(\frac{\nabla\mathcal{L}(\boldsymbol{w})}{\mathcal{L}(\boldsymbol{w})}\right) - \nabla\mathcal{L}_\perp(\mathcal{P}_\perp(\boldsymbol{w}(t)))\right\| \leq \epsilon = \frac{\mu\delta}{100} \leq \frac{1}{100}\|\nabla\mathcal{L}_\perp(\mathcal{P}_\perp(\boldsymbol{w}(t)))\|.
$$

It is easy to verify that $\mathcal{L}_\perp(\cdot)$ is also $L$-smooth in $\mathbb{B}(\boldsymbol{v}^\star; C) - \mathbb{B}(\boldsymbol{v}^\star; \delta)$ for some $L > 0$.

Hence, by setting $\eta = 1/L$, the loss descent has the following lower bound: for any $t \geq T_\epsilon$,

$$
\mathcal{L}_\perp(\mathcal{P}_\perp(\boldsymbol{w}(t))) - \mathcal{L}_\perp^\star = \mathcal{L}_\perp\left(\mathcal{P}_\perp(\boldsymbol{w}(t-1)) - \eta\mathcal{P}_\perp\left(\frac{\nabla\mathcal{L}(\boldsymbol{w}(t-1))}{\mathcal{L}(\boldsymbol{w}(t-1))}\right)\right) - \mathcal{L}_\perp^\star
$$

$$
\leq \mathcal{L}_\perp(\mathcal{P}_\perp(\boldsymbol{w}(t-1))) - \mathcal{L}_\perp^\star - \eta\left\langle\nabla\mathcal{L}_\perp(\mathcal{P}_\perp(\boldsymbol{w}(t-1))), \mathcal{P}_\perp\left(\frac{\nabla\mathcal{L}(\boldsymbol{w}(t-1))}{\mathcal{L}(\boldsymbol{w}(t-1))}\right)\right\rangle
$$

$$
+ \frac{L}{2}\eta^2\left\|\mathcal{P}_\perp\left(\frac{\nabla\mathcal{L}(\boldsymbol{w}(t-1))}{\mathcal{L}(\boldsymbol{w}(t-1))}\right)\right\|^2
$$

$$
\leq \mathcal{L}_\perp \left( \mathcal{P}_\perp(\boldsymbol{w}(t-1)) \right) - \mathcal{L}_\perp^\star - \frac{1}{L} \left( \| \nabla \mathcal{L}_\perp (\mathcal{P}_\perp(\boldsymbol{w}(t-1))) \|^2 - \frac{1}{100} \| \nabla \mathcal{L}_\perp (\mathcal{P}_\perp(\boldsymbol{w}(t-1))) \|^2 \right)
$$

$$
+ \frac{1}{2L} \left( \frac{101}{100} \right)^2 \| \nabla \mathcal{L}_\perp (\mathcal{P}_\perp(\boldsymbol{w}(t-1))) \|^2
$$

$$
\leq \mathcal{L}_\perp \left( \mathcal{P}_\perp(\boldsymbol{w}(t-1)) \right) - \mathcal{L}_\perp^\star - \frac{1}{2L} \cdot \frac{9}{10} \| \nabla \mathcal{L}_\perp (\mathcal{P}_\perp(\boldsymbol{w}(t-1))) \|^2
$$

$$
\leq \mathcal{L}_\perp \left( \mathcal{P}_\perp(\boldsymbol{w}(t-1)) \right) - \mathcal{L}_\perp^\star - \frac{1}{2L} \cdot \frac{9}{10} \cdot 2\mu \left( \mathcal{L}_\perp \left( \mathcal{P}_\perp(\boldsymbol{w}(t-1)) \right) - \mathcal{L}_\perp^\star \right)
$$

$$
\leq \left( 1 - \frac{9\mu}{10L} \right) \left( \mathcal{L}_\perp \left( \mathcal{P}_\perp(\boldsymbol{w}(t-1)) \right) - \mathcal{L}_\perp^\star \right)
$$

$$
\leq \cdots
$$

$$
\leq \left( 1 - \frac{9\mu}{10L} \right)^{t - T_\epsilon} \left( \mathcal{L}_\perp \left( \mathcal{P}_\perp(\boldsymbol{w}(T_\epsilon)) \right) - \mathcal{L}_\perp^\star \right).
$$

Hence, there exists time $t_\epsilon > T_\epsilon$ such that $\mathcal{L}_\perp \left( \mathcal{P}_\perp(\boldsymbol{w}(t_\epsilon)) \right) - \mathcal{L}_\perp^\star < \frac{\mu \delta}{4}$.

On the other hand, the strong convexity implies that

$$
\mathcal{L}_\perp \left( \mathcal{P}_\perp(\boldsymbol{w}(t_\epsilon)) \right) - \mathcal{L}_\perp^\star \geq \frac{\mu}{2} \| \mathcal{P}_\perp(\boldsymbol{w}(t_\epsilon)) - \boldsymbol{v}^\star \| \geq \frac{\mu \delta}{4}.
$$

Thus, we obtain the contradiction.

Step III. There exits $t_k \to \infty$ such that $\mathcal{P}_\perp(\boldsymbol{w}(t_k)) \to \boldsymbol{v}^\star$.

With the help of our proof of Step II, for any $1/k > 0$, there exists $t_k > 0$ such that $\mathcal{P}_\perp(\boldsymbol{w}(t_k)) \in \mathbb{B}(\boldsymbol{v}^\star; 1/k)$. And $t_k \to \infty$ can be ensured simply by setting $T_{\frac{1}{k+1}} > T_{\frac{1}{k}} + 1$ in our proof of Step II.

Step IV. It holds that $\boldsymbol{v}^\star \neq 0$.

If $\boldsymbol{v}^\star = \boldsymbol{0}$, then $\nabla \mathcal{L}_\perp(\boldsymbol{0}) = \boldsymbol{0}$, which implies

$$
\boldsymbol{0} = \frac{1}{|\mathcal{I}|} \sum_{i \in \mathcal{I}} e^0 \mathcal{P}(\boldsymbol{x}_i y_i) = \frac{1}{|\mathcal{I}|} \sum_{i \in \mathcal{I}} \mathcal{P}(\boldsymbol{x}_i y_i).
$$

Therefore,

$$
\frac{1}{|\mathcal{I}|} \sum_{i \in \mathcal{I}} \boldsymbol{x}_i y_i = \frac{1}{|\mathcal{I}|} \sum_{i \in \mathcal{I}} \langle \boldsymbol{x}_i y_i, \boldsymbol{w}^\star \rangle \boldsymbol{w}^\star + \frac{1}{|\mathcal{I}|} \sum_{i \in \mathcal{I}} \mathcal{P}(\boldsymbol{x}_i y_i)
$$

$$
= \frac{1}{|\mathcal{I}|} \sum_{i \in \mathcal{I}} \gamma^\star \boldsymbol{w}^\star = \gamma^\star \boldsymbol{w}^\star,
$$

which is contradict to $\gamma^\star \boldsymbol{w}^\star \neq \frac{1}{|\mathcal{I}|} \sum_{i \in \mathcal{I}} \boldsymbol{x}_i y_i$.

Step V. Final Tightness bound.

From $\mathcal{P}_\perp(\boldsymbol{w}(t_k)) \to \boldsymbol{v}^\star$, there exists $K$ such that $\| \mathcal{P}_\perp(\boldsymbol{w}(t_k)) - \boldsymbol{v}^\star \| \leq \| \boldsymbol{v}^\star \| / 2$, which means

$$
\frac{1}{2} \| \boldsymbol{v}^\star \| \leq \| \mathcal{P}_\perp(\boldsymbol{w}(t_k)) \| \leq \frac{3}{2} \| \boldsymbol{v}^\star \|.
$$

Recalling Theorem C.3, it holds that $\| \boldsymbol{w}(t_k) \| = \Theta(t_k)$.

Then by a direct calculation, we have

$$
\left\| \frac{\boldsymbol{w}(t_k)}{\| \boldsymbol{w}(t_k) \|} - \boldsymbol{w}^\star \right\|^2 = 2 - 2 \frac{\langle \boldsymbol{w}(t_k), \boldsymbol{w}^\star \rangle}{\| \boldsymbol{w}(t_k) \|}
$$

$$
= 2 - 2 \frac{\langle \boldsymbol{w}(t_k), \boldsymbol{w}^\star \rangle}{\sqrt{\langle \boldsymbol{w}(t_k), \boldsymbol{w}^\star \rangle^2 + \| \mathcal{P}_\perp(\boldsymbol{w}(t_k)) \|^2}}
$$

$$= 2 - \frac{2}{\sqrt{1 + \frac{\|\mathcal{P}_\perp(\boldsymbol{w}(t_k))\|^2}{\langle \boldsymbol{w}(t_k), \boldsymbol{w}^\star \rangle^2}}} = \Theta\left(\frac{\|\mathcal{P}_\perp(\boldsymbol{w}(t_k))\|^2}{\langle \boldsymbol{w}(t_k), \boldsymbol{w}^\star \rangle^2}\right)$$

$$= \Theta\left(\frac{\|\mathcal{P}_\perp(\boldsymbol{w}(t_k))\|^2}{\|\boldsymbol{w}(t_k)\|^2 - \|\mathcal{P}_\perp(\boldsymbol{w}(t_k))\|^2}\right) = \Theta\left(\frac{1}{\frac{\|\boldsymbol{w}(t_k)\|^2}{\|\mathcal{P}_\perp(\boldsymbol{w}(t_k))\|^2} - 1}\right) = \Theta\left(\frac{1}{t_k^2}\right),$$

which implies $\left\| \frac{\boldsymbol{w}(t_k)}{\|\boldsymbol{w}(t_k)\|} - \boldsymbol{w}^\star \right\| = \Theta\left(\frac{1}{t_k}\right)$.

Hence, we have proved Theorem 6.4 for NGD.

As for GD, the step size is more mildly, and we only need to repeat the proof of NGD from Step I to Step IV. And in Step V, the only difference from NGD is $\|\boldsymbol{w}(t_k)\| = \Theta(1/\log t_k)$ rather than $\Theta(1/t_k)$ (Theorem C.3), which implies the tightness rate $\Theta(1/\log t_k)$.

$\square$

**Theorem C.3** (Theorem 4.3, (Ji & Telgarsky, 2021)). *Under Assumption 3.1 and 5.4 (ii),*

*(I) (GD). let $\boldsymbol{w}(t)$ be trained by GD (2) with $\eta \leq 1$ starting from $\boldsymbol{w}(0) = \boldsymbol{0}$. Then $\left\| \frac{\boldsymbol{w}(t)}{\|\boldsymbol{w}(t)\|} - \boldsymbol{w}^\star \right\| = \mathcal{O}(1/\log t)$ and $\|\boldsymbol{w}(t)\| = \Theta(\log t)$.*

*(II) (NGD) let $\boldsymbol{w}(t)$ be trained by NGD (3) with $\eta \leq 1$ starting from $\boldsymbol{w}(0) = \boldsymbol{0}$. Then $\left\| \frac{\boldsymbol{w}(t)}{\|\boldsymbol{w}(t)\|} - \boldsymbol{w}^\star \right\| = \mathcal{O}(1/t)$ and $\|\boldsymbol{w}(t)\| = \Theta(t)$.*

**Theorem C.4** (Theorem 4.4, (Ji & Telgarsky, 2021)). *Under the same conditions in Theorem C.3, let $\boldsymbol{w}(t)$ be trained by NGD with $\eta \leq 1$ starting from $\boldsymbol{w} = \boldsymbol{0}$. Then (i) $\mathcal{L}_\perp(\cdot)$ has a unique minimizer $\boldsymbol{v}^\star$ over $\mathrm{span}\{\mathcal{P}_\perp(\boldsymbol{x}_i) : i \in \mathcal{I}\}$; (ii) $\mathcal{L}_\perp(\cdot)$ is strongly convex in any bounded set; (iii) there exists an absolute constant $C > 0$ such that $\|\mathcal{P}_\perp(\boldsymbol{w}(t)) - \boldsymbol{v}^\star\| \leq C$, $\forall t$.*

## D  USEFUL INEQUALITIES

**Lemma D.1.** *(i) For any $x \geq 0$, $\sqrt{1+x} \leq 1 + \frac{x}{2}$; (ii) For any $0 \leq x \leq 1/3$, $\sqrt{1+x} \geq 1 + \frac{x}{3}$.*

**Lemma D.2.** *For a fixed $\gamma \in (0,1)$, consider the function $h(x) = x + 2(1-\gamma^2)\left(\frac{2}{1+e^x} - 1\right), x \in \mathbb{R}$. Then $h'(x) > 0$ holds for any $x \in \mathbb{R}$.*

*Proof of Lemma D.2.*

$$h'(x) = 1 - \frac{4(1-\gamma^2)e^x}{(1+e^x)^2} \geq 1 - \frac{4(1-\gamma^2)e^x}{(2e^{x/2})^2} = \gamma^2 > 0, \ \forall x \in \mathbb{R}.$$

$\square$

**Lemma D.3.** *$\phi(x) = xe^x$ is $e^{-1}$-strongly convex for $x \in [-1,1]$. Hence, $\phi(z_1) \geq \phi(z_2) + \phi'(z_2)(z_1 - z_2) + \frac{1}{2e}(z_1 - z_2)^2$, $\forall z_1, z_2 \in [-1,1]$.*

*Proof of Lemma D.3.* $\phi(x) = xe^x, \phi'(x) = (x+1)e^x, \phi''(x) = (x+2)e^x$. Therefore,

$$\phi''(x) = (x+2)e^x \geq (-1+2)e^{-1} = e^{-1}, \ \forall x \in [-1,1].$$

Furthermore, from the mean value theorem, for any $z_1, z_2 \in [-1,1]$, there exists $\xi \in [-1,1]$ such that

$$\phi(z_1) = \phi(z_2) + \phi'(z_2)(z_1 - z_2) + \frac{\phi''(\xi)}{2}(z_1 - z_2)^2.$$

Thus, $\phi(z_1) \geq \phi(z_2) + \phi'(z_2)(z_1 - z_2) + \frac{1}{2e}(z_1 - z_2)^2$.

$\square$

# E    EXPERIMENTAL SETUPS

In this section, we provide the experiment details in Section 7.

**Experiments on synthetic dataset.**    For synthetic Dataset I, we set $\gamma^\star = \sin(\pi/100)$ and $n = 100$. Then we generate the dataset by setting $\boldsymbol{x}_1 = (\gamma^\star, \sqrt{1 - \gamma^{\star2}})$, $\boldsymbol{x}_2 = (-\gamma^\star, \sqrt{1 - \gamma^{\star2}})$, and generate $\boldsymbol{x}_i \sim \text{Unif}\left(\mathbb{S}^1 \cap \{\boldsymbol{x} : |x_1| \geq \gamma^\star\}\right)$ randomly for $i \geq 3$. As for the label, we set $y_i = \text{sgn}(x_1)$. For synthetic Dataset II, we set $\gamma^\star = \sin(\pi/100)$ and $n = 100$. Then we generate the dataset by setting $\boldsymbol{x}_1 = (\gamma^\star, \sqrt{1 - \gamma^{\star2}})$, $\boldsymbol{x}_2 = (-\gamma^\star, \sqrt{1 - \gamma^{\star2}})$, and generate $\boldsymbol{x}_i \sim \text{Unif}\left(\mathbb{B}(0, 1) \cap \{\boldsymbol{x} : |x_1| \geq \gamma^\star\}\right)$ randomly for $i \geq 3$. As for the label, we also set $y_i = \text{sgn}(x_1)$. For PRGD, we use NGD as the warm-up Phase for 1000 iterations, and then turn it to PRGD.

**Experiments for VGG on CIFAR-10.**    Following (Lyu & Li, 2019), we examine our algorithm for homogenized VGG-16, i.e., the bias term only exists in the input layer. Additionally, in this setting, we employ mini-batch gradient descent instead of the full gradient, and we need to fine-tune the learning rate of NGD and PRGD. Both of these two algorithms share the same learning rate scheduling strategy as described in (Lyu & Li, 2019).

