# OpenReview forum: "Achieving Margin Maximization Exponentially Fast via Progressive Norm Rescaling"
_ICLR.cc/2024/Conference — ICLR 2024 Conference Withdrawn Submission_

### Official Review · Reviewer_uCJS · 2023-10-23

**Soundness:** 3 good
**Presentation:** 2 fair
**Contribution:** 2 fair
**Rating:** 3
**Confidence:** 4

**Summary:**

This paper introduces a new gradient-descent-based algorithm that achieves faster convergence rate on separable linear classification problems. The authors constructed examples which illustrate the limitations of vanilla GD and normalized GD. With such examples in mind, the authors proposed a "progressive rescaling" approach to more precisely control the iterates' orthogonal component to the maximum-margin solution. This paper proves that the proposed "Progressive Rescaling GD" (PRGD) converges to the maximum-margin solution exponentially quickly and the numerical experiments show that PRGD also achieves better convergence rate than GD on a variety of synthetic and real-world datasets.

**Strengths:**

I very much like two specific ideas in this paper:
1. The PRGD algorithm is quite clever. While it is understood that the rate of convergence on logistic regression depends on the magnitude of the iterates (e.g. [1] and [2]), I am not aware of any accelerated algorithms in the literature that directly control this quantity. This paper neatly extended such simple observation by separately consider the components parallel and orthogonal to the max-margin solution. I am pleasantly surprised that the resulting algorithm is simple and its associated proofs do not require much heavy machinery.

2. The lower bounds given by Theorem 6.4 are valuable contribution to the literature. I believe this result can offer useful insights for future studies in the implicit bias and convergence of GD-based optimization algorithms.


[1] Nacson, Mor Shpigel, et al. "Convergence of gradient descent on separable data." The 22nd International Conference on Artificial Intelligence and Statistics. PMLR, 2019.

[2] Sun, Haoyuan, et al. "Mirror descent maximizes generalized margin and can be implemented efficiently." Advances in Neural Information Processing Systems 35, 2022.

**Weaknesses:**

Despite the contributions of this paper (as I noted above), I feel that there are some significant gaps in this paper.

1. Assumption 5.4 is not consistent with the definition of non-degenerate data in [3]. In Soudry et al. [3], "non-degenerate data" refers to the case where the support vectors are linearly independent, but this paper considers the case where the span of support vectors is equal to the span of all data vectors. Therefore, this brings major concerns on whether Assumption 5.4 is reasonable. I am inclined to say no, as this assumption does not hold in an over-parameterized setting where $d \gg n$.

1b. There are some additional issues with the author's citation of [3]. First, [3] has a separate result showing convergence in their version of degenerate case. Thus, referencing [3] is not a valid justification for making Assumption 5.4 even if it were consistent with the literature. Also, there is no Theorem 4.4 in [3], so the authors were looking at the wrong version of the paper.

2. Theorem 6.2 mentioned the choice of parameter $R_k = \exp(\Theta(k))$. In contrast, its proof uses $R_k = \frac{D \lVert w(T_k) \rVert}{\lVert \mathcal{P}_\perp(w(T_k)) \rVert}$. While the proof does show that these quantities are equal, it does raise the question of whether the choice of $R_k$ is practical, as we do not know $w^\star$ ahead of time and therefore cannot compute the denominator. The authors need to present a choice of $R_k$ that is both *explicit* and *computable*.

3. The proofs of both Corollary 6.3 and Theorem 6.4 are incomplete.

4. The experimental section is very lacking. First, text in the figures are too small. Secondly, the figure captions are too short; for example, I spendt several minutes trying to figure out what are the left plot in Figures 2a and 2b. Next, the authors did not give their choice of hyper-parameters. Finally, and only a minor criticism on the experiments, the deep neural net experiment is a bit thin, as VGG is a not a well-performing architecture, even for a small dataset like CIFAR-10.


Finally, some minor comments:

5. There are a lot of typos in the paper. For example, "we have" is redundant in the first bullet point of Theorem 6.4. Also, in Appendix B, the proof should be for Theorem 5.5, not 5.4.

6. I suggest adding a brief discussion of Theorem 6.4 in Section 4, as it is a substantial result but barely mentioned outside of Section 6.2.

7. In Definition 5.3, the term "semi-cylindrical surface" is misleading, as this normally implies a half-cylinder cut along the direction parallel to its axis, NOT along the cross section.

8. The proofs can be better structured. There are several auxiliary lemmas (e.g. Lemma A.3 and A.4) that are not formally stated until the end of the section. They are preferably introduced in a separate subsection before the main proofs.

[3] Soudry, Daniel, et al. "The implicit bias of gradient descent on separable data." The Journal of Machine Learning Research 19.1, 2018.

**Questions:**

I wrote down all of my concerns in the Weaknesses section. In particular, my first and second points are the two most significant issues that I found.

---

### Official Review · Reviewer_wv8x · 2023-11-01

**Soundness:** 3 good
**Presentation:** 3 good
**Contribution:** 3 good
**Rating:** 6
**Confidence:** 3

**Summary:**

The paper analyzes why gradient descent and normalized gradient descent cannot efficiently maximize margins. A critical component of their analysis is a term they define as centripetal velocity, the inner product between the normalized loss gradient and the normalized projected component of the weight vector. Algorithms that maintain a larger centripetal velocity maximize margins more efficiently, and thus their proposed algorithm PRGD actively tries to have a large centripetal velocity.

**Strengths:**

Proposition 4.1 shows clear separation between PRGD and NGD in terms margin maximization on a specific dataset.

For general linearly separable datasets, theorem 6.4 shows that the sequence of weights produced by GD and NGD have subsequences that have margin maximization rates $\Theta(1/ \ln t)$ and  $\Theta(1/ t)$ respectively.

Using centripetal velocity to explicitly show the inadequacies of GD and NGD for maximizing margin is novel. The notion of "favorable" semi-cylindrical surfaces is interesting and provides a concrete way to improve upon GD and NGD.

**Weaknesses:**

Theorem 6.2 (theorem showing that PRGD maximizes margin at an exponential rate) uses the fact that the PRGD algorithm has access to the diameter and height of the "favorable"-semi cylindrical surface and $\gamma$. Besides the issue that such values are not given in practice, in this setting, PRGD has an advantage over GD and NGD in that it can choose some of its hyper parameters (progressive radius) after the problem is chosen.

**Questions:**

In theorem 6.4, the results show the existence of a subsequence of weights produced by NGD and GD that have a slow margin maximization rate. Is there a dataset where if one instead considers the sequence $w'_t$ where
$$ w'_t \in \arg\min_{s \leq t} \| \frac{ w*(s) }{ \| w*{s}\| } - w^*} \| $$ has a much better margin maximization rate?


In the experiments, when running PRGD on $nonlinear$ datasets, was there directional convergence (if so did directional convergence
occur at a faster rate than models trained with NGD / GD)?

What would be the analogue of favorable semi-cylindrical surfaces in nonlinear datasets (namely what sort of central directions would one consider)?

---

### Official Review · Reviewer_WCCZ · 2023-11-03

**Soundness:** 3 good
**Presentation:** 3 good
**Contribution:** 2 fair
**Rating:** 5
**Confidence:** 4

**Summary:**

This paper develops a new algorithm for the logistic regression problem, which can provably achieve much faster directional convergence compared to GD and NGD. The key idea is to progressively rescale the model parameter at a proper rate to maintain a non-degenerate centripetal velocity. Finally, the margin maximization rate can be improved.

**Strengths:**

* This paper proposes a new optimization algorithm for logistic regression that can achieve a faster directional convergence rate.
* This paper points out a new property called centripetal velocity, which can be used in studying the implicit bias of other models such as homogeneous models.

**Weaknesses:**

* The scope of this paper is a bit narrow, it only focuses on linear models for separable data. Besides, instead of proving faster directional convergence, there are no new results compared to the previous work.

* Even for the directional convergence, I do not see significant impacts of achieving faster direction convergence to the generalization. In particular, the model that achieves $\gamma/2$ margin can already attain good generalization performance, we do not need to push the margin to be very close to $\gamma$.

* Regarding the PRGD algorithm, I am not clear whether this algorithm can be generalized to other settings. Using exponentially increasing scaling may be weird in practice. In fact, the original implicit bias work for logistic regression is nice as it reveals the key properties of the commonly used GD algorithm,   while crafting (potentially impractical) algorithms for achieving faster directional convergence may be incremental.

**Questions:**

* Does the algorithm need to use the information of the margin?
* What's the convergence when $R_k$ is not exponential in $k$?
* In Section 7.2, what are the choices of $R_k$ and $T_k$?